# Scalable nanoscale positioning of highly coherent color centers in prefabricated diamond nanostructures

Sunghoon Kim[1], Paz London[1], Daipeng Yang[1], Lillian B. Hughes [1], Jeffrey Ahlers [1], Simon Meynell[1], William J. Mitchell [2], Kunal Mukherjee[3] & Ania C. Bleszynski Jayich [1] ✉

Nanophotonic devices in color center-containing hosts provide efficient readout, control, and entanglement of the embedded emitters. Yet control over color center formation – in number, position, and coherence – in nanophotonic devices remains a challenge to scalability. Here, we report a controlled creation of highly coherent diamond nitrogen-vacancy (NV) centers with nanoscale three-dimensional localization in prefabricated nanostructures with high yield. Combining nitrogen $\delta$-doping during chemical vapor deposition diamond growth and localized electron irradiation, we form shallow NVs registered to the center of diamond nanopillars with wide tunability over NV number. We report a positioning precision of ~4 nm in depth and 46(1) nm laterally in 280 nm-diameter pillars (102(2) nm in bulk diamond). We reliably form single NV centers with long spin coherence times (average $T_2^{Hahn} = 98\,\mu s$) and higher average photoluminescence compared to NV centers randomly positioned in pillars. Our method can improve the performance of various NV-based devices. In the realm of magnetic sensing, we achieve a $3 \times$ improved yield of NV centers with single electron-spin sensitivity over conventional implantation-based methods. Our high-yield defect creation method will enable scalable production of solid-state defect sensors and processors.

Optically addressable solid-state spin defects are versatile tools for quantum-enhanced technologies[1,2]. The photonic degree of freedom enables single-spin readout[3] and control[4,5] and entanglement generation[6,7]. Moreover, engineered nanophotonic structures can greatly enhance spin-photon interfaces, where customized structures such as cavities[8–13], solid immersion lenses[14], metalenses[15], nanobeams[10,16] or nanowires[17,18] can be fabricated in the host material to increase collection efficiency[19], waveguide emitted photons[9] or Purcell-enhance photon emission[20–25]. In particular, diamond color centers are readily interfaced with engineered photonic structures to provide these advanced functionalities[26–29]. To realize efficient defect-photon interfaces, it is necessary to engineer a good spatial overlap

between the optical mode of the nanostructured device and the defect. However, control over the formation of color centers in position and number, while maintaining reproducibly long spin coherence, remains an outstanding problem in realizing scalable fabrication of devices equipped with quantum-enhanced functionalities.

Conventionally, color centers are created via ion implantation prior to device fabrication. The implantation dosage is chosen to match a target number of defects per device, but placement is random in the nanostructures. Subsequently, in a time-intensive, low-yield, and hence nonscalable post-selection process, devices with defects at the ideal position (e.g., at the mode maximum of the optical field) are selected. Moreover, the properties of the selected defects can degrade

[1]Department of Physics, University of California, Santa Barbara, Santa Barbara, CA, USA. [2]Nanofabrication Facility, University of California, Santa Barbara, Santa Barbara, CA, USA. [3]Department of Materials Science and Engineering, Stanford University, Palo Alto, CA, USA. ✉e-mail: ania@physics.ucsb.edu

during the subsequent device fabrication process, e.g., at the etching step[30]. Alternate approaches utilize highly specialized, home-built localized implantation techniques to spatially co-locate a defect and a nanostructure. Atomic force microscopy-assisted implantation[31,32] and focused ion beam implantation[33] have demonstrated lateral confinement inside prefabricated nanostructures to ~20–30 nm. Recently, a patterning technique[34] involving implantation masks has shown ~15 nm lateral positioning precision in a nanopillar, though the technique is limited to nanopillar geometries.

However, implantation-based techniques have several drawbacks. Most critically, they suffer from collateral damage incurred during implantation (e.g., vacancy clusters) that adversely affect optical[35,36] and spin properties[34,37,38] of nitrogen-vacancy (NV) centers. For instance, spin coherence times have been limited to $T_2^{Hahn}$ less than 20 and 50 μs for >90% of 10 and 15 keV implanted NV centers, respectively[34,37]. Damage is also exacerbated at higher implantation dosages[38], which are necessary for, e.g., achieving high defect densities or ensuring the presence of a defect in a small target volume. Further, the spread in the depth of implanted defects hinders their precise vertical positioning. For example, 15 keV implanted NV centers have a vertical spread of >14 nm due to implantation straggling[39] and ion

channeling[40], effects that become even more severe at higher implantation energies.

In contrast to ion implantation, nitrogen δ-doping during chemical vapor deposition (CVD) diamond growth enables NV formation with reproducibly long spin coherence, nanometer-scale depth confinement even at large depths, and independent tunability over a wide range of nitrogen and NV densities[41–45]. Previous studies on δ-doped CVD-grown diamond demonstrated NV center densities tunable from <1 to 47 ppm nm using electron irradiation[41,42,46], <3 nm depth confinement, and reproducibly long coherence times, with even 15 nm-deep NVs showing $T_2^{Hahn} > 100 μs$[47]. Depth confinement of NV centers using δ-doping has been used to enhance their coupling to nanophotonic devices, such as photonic crystal nanobeam cavities[48], but without controlled lateral positioning. Local vacancy creation techniques[49–54] can also provide lateral confinement in addition to the depth confinement afforded by δ-doping, but this capability has only been demonstrated on a bulk substrate without alignment to prefabricated photonic structures[44,46].

In this paper, we demonstrate high-throughput, localized formation of highly coherent NV centers aligned to prefabricated nanophotonic structures, with applications in, e.g., nanoscale magnetic sensing and photon-driven entanglement of solid-state emitters. In a process we refer to as δ-electron irradiation, we register a 200 keV electron beam with 20 nm spot size to the center of a diamond nanopillar; the nanopillar is fabricated in CVD-grown diamond with a 53 nm-deep, δ-doped nitrogen layer. By controlling the electron dose and annealing time, we tune the average number of NVs per irradiation spot inside nanopillars from ~0 to 10. We report lateral confinement of created NVs to a standard deviation of 102(2) nm in unpatterned diamond in addition to ~4 nm vertical confinement. Lateral confinement is improved to 46(1) nm in 280 nm diameter pillars and 72(1) nm in 480 nm diameter pillars. We find that our observations agree well with Monte Carlo (MC) simulations based on a simple diffusion-capture model. Importantly, the NVs formed using our method feature repeatably long spin coherence time (average $T_2^{Hahn} = 98(37) μs$) with a high spin-dependent photoluminescence (PL) contrast of 18 (4)%. Additionally, we observe a 1.8× enhancement of PL from NV centers localized to pillars compared to pillars with non-localized NVs. For nanoscale magnetic sensing, we demonstrate ×3 increase in the expected yield of single electron spin-sensitive magnetometers compared to conventional methods. Our method is readily applicable to the formation of improved, emitter-containing devices in a variety of application spaces as well as device geometries beyond the nanopillars used here. Overall, this technique facilitates the scalable fabrication of state-of-the-art solid-state defect-assisted devices, where scalable refers to a high-yield, time-efficient process that leverages commercially available tools.

## Results

### Targeted formation of NVs in nanostructures

Our targeted formation of NVs in prefabricated nanostructure utilizes localized electron irradiation and timed vacuum annealing, as shown schematically in Fig. 1a, b. We first fabricate our device from a CVD-grown diamond with a 53 nm-deep $^{15}N$ δ-doped layer, as described in "Methods"[41]. We use electron beam lithography (EBL) followed by inductively coupled plasma-reactive ion etching (ICP-RIE) recipes to transfer nanostructures onto the diamond substrate. We use a negative electron beam resist (FOx-16, Dow Corning) as a mask for etching ≈1 μm tall features with Ar/O₂ plasma. Nanopillars with diameters 280 nm and 480 nm, square mesas with 20 μm × 20 μm dimensions, and alignment marks were fabricated, as shown in the scanning electron micrograph (SEM) in Fig. 1c.

To laterally localize NV centers, we use a commercially available 200 keV EBL tool (JBX-8100FS, JEOL Ltd.) to δ-electron irradiate the centers of the nanopillars using an electron beam of 20 nm spot size

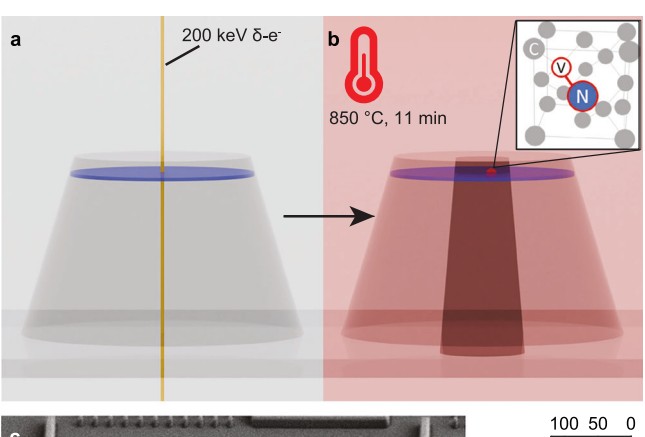

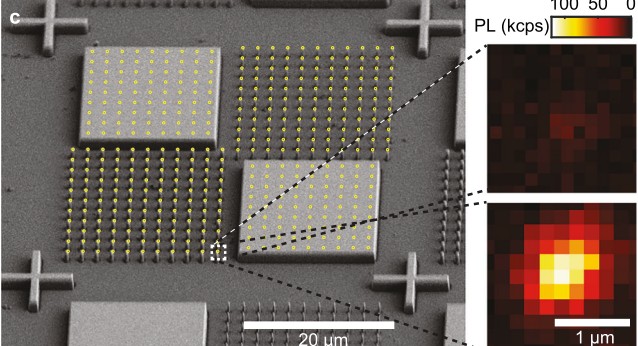

**Fig. 1 | Targeted formation of color centers aligned to prefabricated diamond nanostructures. a** Schematic of our formation method showing a nanopillar containing a near-surface nitrogen δ-doped layer (blue disk). In δ-electron irradiation, an electron beam of 20 nm spot size (yellow line) irradiates the center of a pillar, creating vacancies along its trajectory. **b** Upon annealing, monovacancies diffuse to form a vacancy-rich region (dark shaded region) in which they can be captured by nitrogen atoms to form NV centers (inset). **c** A scanning electron micrograph of a unit block of etched diamond pillars and mesas framed by alignment marks. Each $50 \times 50\ \mu m^2$ unit block consists of two unetched mesas (top left and bottom right), and two regions of nanopillars with a diameter of 280 nm (top right) and 480 nm (bottom left). The overlaid yellow circles denote the target positions of the electron beam (scale bar: 20 μm). The insets show confocal PL images of a single 480 nm pillar under 170 μW of 532 nm excitation before (top) and after (bottom) $4.8 \times 10^{21}\ e^-/cm^2$ δ-electron irradiation and annealing (scale bar: 1 μm).

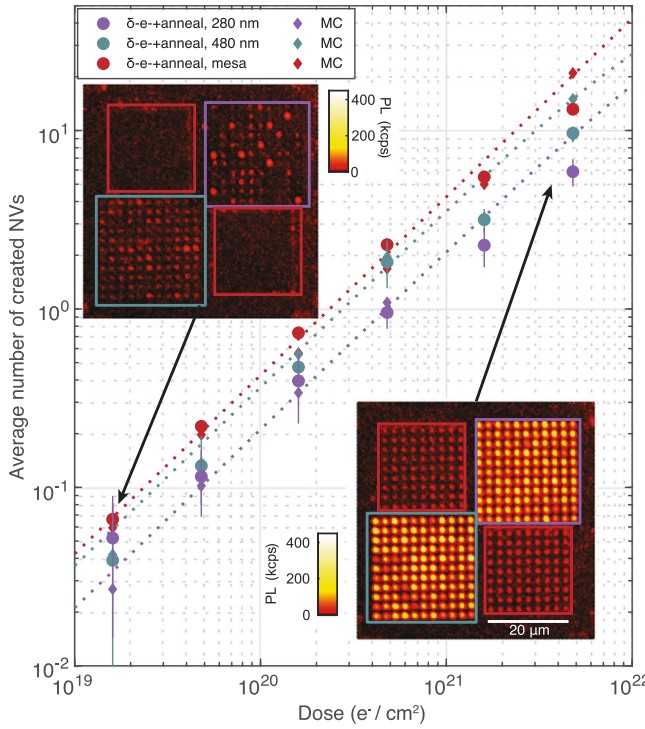

**Fig. 2 | Electron dose control over NV creation.** Plotted are the average number of created NVs per spot in 280 nm diameter pillars (purple circles), in 480 nm diameter pillars (teal circles), and in the mesas (red circles). The error bars denote 95% confidence interval of the NV number estimation. The results from MC simulations (diamonds) are plotted in corresponding colors and show good agreement with the measurements. The fitted curves for the MC simulation results (dotted lines) are shown as a guide to the eye. The insets show confocal micrographs of a unit block after irradiation and annealing with a dose of $1.6 \times 10^{19}$ e$^-$/cm$^2$ (top left) and $4.8 \times 10^{21}$ e$^-$/cm$^2$ (bottom right). (scale bar: 20 μm). The overlaid boxes indicate the locations of the 280 nm pillars (purple box), 480 nm pillars (teal box), and mesas (red boxes).

(Fig. 1a); these electrons can penetrate into the diamond and displace carbon atoms along their trajectory up to ~50 μm below the surface[46,55], creating a narrow pencil of vacancies (Fig. 1a, see Supplementary Section 3). We note that 145 keV is the threshold energy for vacancy creation in diamond[46] and only recently have commercial EBL tools exceeded 150 kV. The resulting monovacancy density depends on electron dose, which we tune from $1.6 \times 10^{19}$ e$^-$/cm$^2$ to $4.8 \times 10^{21}$ e$^-$/cm$^2$ by adjusting the dwell time while keeping the beam current constant at 20 nA. Subsequent annealing at 850 °C for 11 min in vacuum promotes the diffusion of monovacancies (Fig. 1b). When a monovacancy diffuses to a site adjacent to a nitrogen atom, it can get captured to form an NV center (Fig. 1b, inset). We note that NVs formed in the $\delta$-doped N-layer can be identified by their $^{15}$NV hyperfine structure, revealed with pulsed electron spin resonance (ESR) spectroscopy. Then, the device is cleaned in a boiling nitrating acid (1:1 HNO$_3$:H$_2$SO$_4$) and annealed at 450 °C in air for 4 hours. All NV measurements were taken using a home-built confocal microscope with 532 nm excitation[56] (see "Methods").

First, we show control over the number of NVs formed per nanopillar by varying the $\delta$-electron irradiation dosage. In Fig. 2, we sweep the irradiation dose from $1.6 \times 10^{19}$ e$^-$/cm$^2$ to $4.8 \times 10^{21}$ e$^-$/cm$^2$ and measure the average number of NVs created in 280 nm (purple circles) and 480 nm (teal circles) diameter pillars. We use a maximum likelihood estimation method[46] based on continuous wave-ESR spectroscopy (CW-ESR) taken on 121 pillars for each pillar size and irradiation dose after annealing (see Supplementary Section 1). With increasing irradiation dose, the average number of created NVs

increases monotonically from ~$5 \times 10^{-2}$ to 5.9(7) and 9.7(4) for 280 nm and 480 nm diameter pillars, respectively. We subtract the contribution from as-grown NVs, characterized by measuring non-irradiated pillars of each diameter (see Supplementary Section 1). We also estimate the average NV number per spot in $\delta$-electron irradiated mesas by measuring the total PL around the target areas normalized by that of single NVs, as plotted in red circles in Fig. 2. Likewise, we observe a monotonic increase in the NV formation with irradiation dose.

The increase in NV number with pillar diameter indicates that vacancies diffuse at least as far as the radius of the smaller pillar, and comparing our results to MC simulations ("Methods") of a simple diffusion-capture model of NV formation, we extract a monovacancy diffusion constant $D_V = 17(4)$ nm$^2$ s$^{-1}$ (see Supplementary Section 5). Simulation results are plotted as diamonds in Fig. 2. At higher irradiation doses ($\geq 1.6 \times 10^{21}$ e$^-$/cm$^2$), simulations slightly overestimate NV number, which we attribute to the creation of vacancy clusters at high monovacancy density[57–59], giving rise to sublinear monovacancy creation efficiency. In the next section, we characterize $D_V$ via an alternate approach, arriving at a similar value, and we further discuss the results.

We identify ~$5 \times 10^{20}$ e$^-$/cm$^2$ and $3 \times 10^{20}$ e$^-$/cm$^2$ as the target dose for an average of 1 NV per 280 nm and 480 nm pillar, respectively, facilitating the fabrication of devices based on single isolated defects for sensing and networking applications. We expect the target electron dose to vary for different nitrogen densities and nanostructure geometries.

We next demonstrate high spatial confinement of NV centers aligned to diamond nanostructures, which is necessary for optimizing spatial overlap between the defect qubit and the structure's photonic modes. In Fig. 3, we quantify the lateral positioning precision of created NVs afforded by $\delta$-electron irradiation. To do so, we first investigate arrays of NVs patterned in $20 \times 20$ μm$^2$ mesas and quantify the deviation $\sigma_{loc}$ of NV positions from their target irradiation spot. The irradiation pattern on the mesa is shown in yellow circles in Fig. 1c and example confocal images are shown in Fig. 2 (bottom right and top left square of each inset). We use these featureless mesas to avoid exciting photonic modes of the nanopillars that modify the NV emission pattern and obfuscate the actual NV position. We estimate $\sigma_{loc}$, the standard deviation of lateral NV positioning in the mesas, by pixel-wise averaging 162 tiles in the confocal image, with each $2 \times 2$ μm$^2$ tile centered on a single irradiation spot; the tiles were obtained by cutting a ~$40 \times 40$ μm$^2$ confocal image containing two mesas (the area shown in Fig. 1c) into a regular grid (see Supplementary Section 6). Prior to this image cutting we apply global affine transformations to the original confocal image to account for optical aberrations in our imaging system. We then repeat this procedure over several different $40 \times 40$ μm$^2$ confocal image areas to arrive at the pixel-wise averaged confocal images shown in the insets of Fig. 3b. Each of these images has a finite lateral spread $\sigma_{tot}$ with respect to the target positions due to three contributions: the lateral NV positioning precision of our patterning technique $\sigma_{loc}$, the point spread function (PSF) of our imaging system $\sigma_{PSF}$, and residual global aberrations not removed after the first set of global affine transformations $\sigma_{sys}$:

$$\sigma_{tot}^2 = \sigma_{loc}^2 + \sigma_{PSF}^2 + \sigma_{sys}^2. \tag{1}$$

To extract $\sigma_{loc}$ we first measure $\sigma_{tot}$ by fitting the averaged confocal images with a 2D Gaussian curve,

$$PL(x, y) = PL_{max} e^{-(x^2 + y^2)/(2\sigma_{tot}^2)}, \tag{2}$$

where PL$_{max}$ is the maximum PL of the averaged confocal image. The radial profiles of the averaged images (circles) and the fits (solid lines) are shown in Fig. 3b. Then, we measure $\sigma_{PSF} = 235$ nm by imaging six single NVs in the mesa region and individually fitting them to a 2D

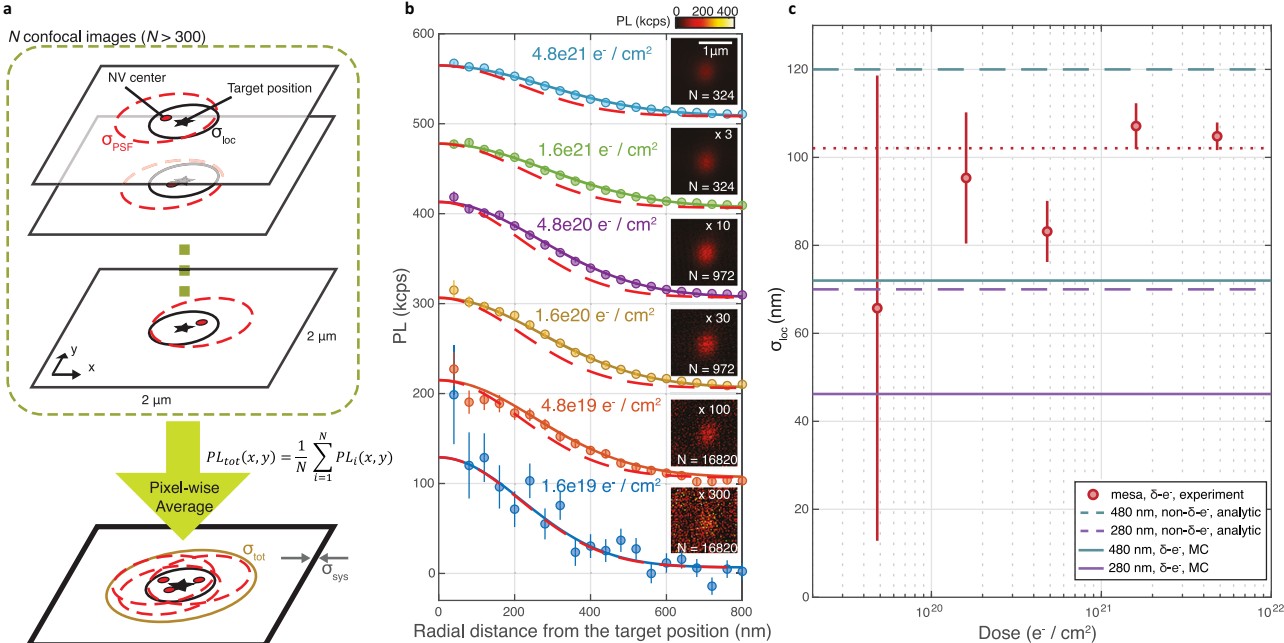

**Fig. 3 | Quantifying spatial confinement of formed NVs. a** Schematic of pixel-wise averaging method for estimating $\sigma_{loc}$. NVs are positioned at the target position (black star) with a lateral precision $\sigma_{loc}$ (black solid line). Red dashed line indicates the point spread function $\sigma_{PSF}$ of our confocal microscope. $N$ confocal images of unpatterned mesas are pixel-wise averaged to give a $\sigma_{tot}$ from which $\sigma_{loc}$ is extracted as discussed in the main text. Residual optical aberrations are indicated by $\sigma_{sys}$. **b** Data points show the radial PL profiles (averaged over angle) of the pixel-wise averaged images (bin size: 40 nm). Error bars show standard error of the averaging. Colored solid lines are 2D Gaussian fits, from which $\sigma_{tot}$ is extracted. For comparison, red dashed lines show the radial cuts of the 2D Gaussian functions with a peak width of $\sqrt{\sigma_{PSF}^2 + \sigma_{sys}^2}$. Plots are offset for clarity, each with a relative $\Delta PL$ of 100 kcps. Insets show the averaged images with PL scaling inversely proportional to the dose (scale bar: 1 μm). **c** Measured $\sigma_{loc}$ in the mesas (red circles, error bar: 95% confidence interval) is plotted as a function of dose. The error bars increase at lower doses due to the smaller number of measured NVs per spot, and the $1.6 \times 10^{19}$ e$^-$/cm$^2$ dose data point is omitted because of large errorbars >1 μm. The red dotted line indicates the average $\sigma_{loc}$ of 102(2) nm measured in the mesas. Solid lines are MC simulations of $\sigma_{loc}^{pillar}$ in δ-e$^-$-irradiated pillars, which are lower than the analytically calculated $\sigma_{loc}^{pillar}$ for NVs created without localization methods (dashed lines).

Gaussian function $PL_{single}(x,y) \propto e^{-(x^2+y^2)/(2\sigma_{PSF}^2)}$ (see Supplementary Section 6). We characterize the residual global aberration of our transformed images to be $\sigma_{sys} = 41$ nm (see Supplementary Section 6). Finally, we extract $\sigma_{loc}$ for the different irradiation dosages and plot the results in Fig. 3c (red circles). The data shows minimal dependence of $\sigma_{loc}$ on dose, with an average $\sigma_{loc} = 102(2)$ nm (red dotted line).

Since δ-electron irradiation creates highly localized mono-vacancies (see Fig. 1 and Supplementary Section 3), we attribute the majority of $\sigma_{loc}$ to vacancy diffusion during annealing. Hence our measured $\sigma_{loc}$ provides an estimate of $D_V = 21$ nm$^2$ s$^{-1}$ via comparison with MC simulations (see Supplementary Section 7). This number is consistent with the estimated $D_V = 17(4)$ nm$^2$ s$^{-1}$ from NV number measurements (Fig. 2) and is within the range of values reported in the literature. We note that reported values of $D_V$ show strong sensitivity to experimental conditions including annealing temperature, annealing time, and vacancy creation method[44,46,60–62]. For instance, ref. 44 measured $D_V = 6.5$ nm$^2$ s$^{-1}$ at 850 °C-30 min annealing and ref. 61 found $D_V \leq 40$ nm$^2$ s$^{-1}$ at 1050 °C-2 h annealing. Further, ref. 62 found $D_V$ can be enhanced due to transient dynamics during the first few minutes of annealing ($D_V \sim 300$ nm$^2$ s$^{-1}$ for 2 min annealing at 1000 °C).

We next estimate the lateral NV positioning precision in the nanopillars $\sigma_{loc}^{pillar}$. We use MC simulations because a direct measurement is challenging due to the effect of the pillars' photonic modes on the confocal images. As shown in Fig. 3c, $\sigma_{loc}^{pillar}$ (solid lines) is smaller than $\sigma_{loc}$, and notably smaller than $\sigma_{loc}^{pillar}$ for NVs uniformly distributed in a nitrogen layer bounded by the pillar walls (where $\sigma_{loc}^{pillar} = 1/4 \times$ pillar diameter), as would result from the conventional method of forming pillars after NV formation. In contrast, our technique achieves improved lateral confinement even in pillars with comparable size to $\sigma_{loc}$, a fact we attribute to vacancy absorption at the pillar sidewalls during annealing. Specifically, we find $\sigma_{loc}^{pillar} = 46(1)$ nm and 72(1) nm in our δ-electron irradiated 280 nm and 480 nm pillars. This improved lateral confinement is expected to enhance the coupling to the nanopillar photonic mode, as shown in the next section.

## Spin and optical properties of single NVs in nanopillars

We next present the spin coherence and photoluminescence properties of single $^{15}$NV centers created inside nanopillars, confirmed by second-order correlation measurements (see Supplementary Section 8) and hyperfine-resolved pulsed ESR measurements. Figure 4a shows a histogram of the Hahn echo coherence time, $T_2^{Hahn}$, of 12 NV centers formed by $1.6 \times 10^{20}$ e$^-$/cm$^2$-irradiation. We observe reliably long $T_2^{Hahn}$ with a mean of 98(37) μs, which we attribute to the gentle nature of our NV formation process, producing little collateral damage that can adversely affect coherence. The coherence time is consistent with the limit imposed by the surrounding substitutional nitrogen (P1 center) bath(see Supplementary Section 9). In contrast, 15 keV ion implantation results in NVs with the majority (>90%) having $T_2^{Hahn}$ less than 50 μs[34]. As shown in the inset, we also measure long $T_2^{Hahn}$ for single NV pillars with $4.8 \times 10^{19}$ e$^-$/cm$^2$ irradiation, while we see a reduced $T_2^{Hahn}$ of 36(8) μs for $4.8 \times 10^{20}$ e$^-$/cm$^2$ (Fig. 4a, inset). We attribute this reduction to increased vacancy-related damage, consistent with previous reports[41,60].

We also demonstrate favorable optical properties of our δ-electron irradiated single NV centers, namely good spin-dependent optical readout contrast and high photon collection rates. Figure 4b shows an average spin-dependent Rabi PL contrast $C_{Rabi} = \frac{PL_0 - PL_{\pm 1}}{PL_0}$ of 18(4)% at $1.6 \times 10^{20}$ e$^-$/cm$^2$, where $PL_0$ and $PL_{\pm 1}$ are the PL for NV electronic spin states 0 and ±1, respectively. The inset shows the dependence of $C_{Rabi}$

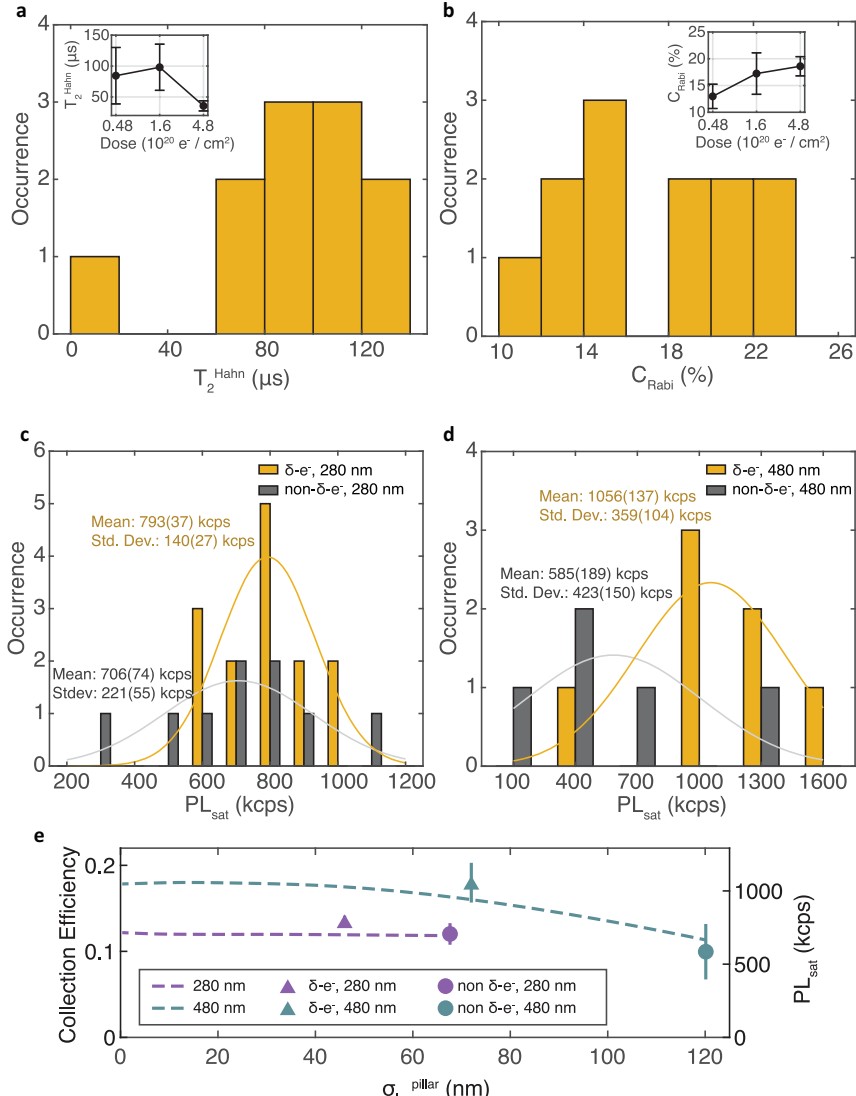

**Fig. 4 | Spin coherence time $T_2^{Hahn}$ and PL properties of single NVs formed in nanopillars. a** Histogram of $T_2^{Hahn}$ for NVs $\delta$-electron irradiated with $1.6 \times 10^{20}$ e⁻/cm² in both 280 nm and 480 nm diameter pillars. (Inset) Average $T_2^{Hahn}$ as a function of irradiation dose. **b** Histogram of Rabi contrast $C_{Rabi}$ at $1.6 \times 10^{20}$ e⁻/cm². (Inset) Average $C_{Rabi}$ as a function of $\delta$-electron irradiation dose. Histogram of $PL_{sat}$ in **c** 280 nm and **d** 480 nm pillars. Each plot shows both non-irradiated (gray) and $1.6 \times 10^{20}$ e⁻/cm²·$\delta$-electron irradiated (yellow) pillars. The solid curves are the Gaussian fit for the histograms. **e** Dashed lines indicate mean photon collection efficiency calculated from FDTD simulations for a given lateral distribution $\sigma_{loc}^{pillar}$. Data points are the experimentally measured sample mean of the $PL_{sat}$ distribution for $\delta$-electron irradiated pillars (triangles) and non-irradiated pillars (circles) for 480 nm (teal) and 280 nm (purple) diameter pillars. The error bars denote the standard error of the estimation of the population mean. The limits of the secondary y-axis are chosen so that the simulated collection efficiency and measured $PL_{sat}$ for the nonirradiated pillars line up.

on irradiation dose with no evidence of reduced contrast at the highest dosages compatible with single NV formation.

In Fig. 4c, d, we show histograms (yellow) of the saturation count rate $PL_{sat}$ (with background PL subtracted) for 280 nm and 480 nm pillars, respectively (see Supplementary Section 10). The means of the two $PL_{sat}$ histograms are 0.793(37) Mcps and 1.056(137) Mcps, respectively. Also plotted in gray are the $PL_{sat}$ histograms of non-irradiated pillars with as-grown single ¹⁵NVs, which have a uniform spatial distribution inside the pillars. The non-irradiated pillars show a lower mean $PL_{sat}$ compared to the $\delta$-electron irradiated pillars by a factor of 1.8 and 1.1 for 480 nm and 280 nm pillars, respectively. The increase in mean $PL_{sat}$ in the $\delta$-electron irradiated 480 nm pillars is statistically significant. Higher $PL_{sat}$ values could be achieved with pillar geometries optimized for increased collection efficiency, such as parabolic pillars[63] or tapered pillars [34], where with pillars with a ~70° sidewall angle, the authors achieved an average $PL_{sat}$ = 1.6 Mcps.

In Fig. 4e, we conduct finite-difference time-domain (FDTD) simulations to study the effect of lateral localization on $PL_{sat}$ in nanopillars (see "Methods"). The simulations (dashed lines) and the data are in good agreement, indicating that increased localization precision is the main contributor to increased $PL_{sat}$ for the 480 nm pillars, while $PL_{sat}$ does not depend strongly on $\sigma_{loc}^{pillar}$ in the 280 nm pillars. This can be attributed to a higher-order waveguiding mode, TE01, strongly outcoupling photons from NVs close to the edge of 280 nm pillars. In contrast, for 480 nm pillars, NVs close to the edge are weakly coupled to the TE01 mode, while NVs close to the center are coupled to both the fundamental mode HE11 and the higher order mode HE12, thus exhibiting sharper PL dependence in NV lateral displacement. For both pillars, our $\sigma_{loc}^{pillar}$ nearly maximizes photon collection efficiency. For future applications, color center localization should be performed in conjunction with FDTD simulations to optimize the emitter overlap with the photonic mode. Overall, high collection efficiency combined

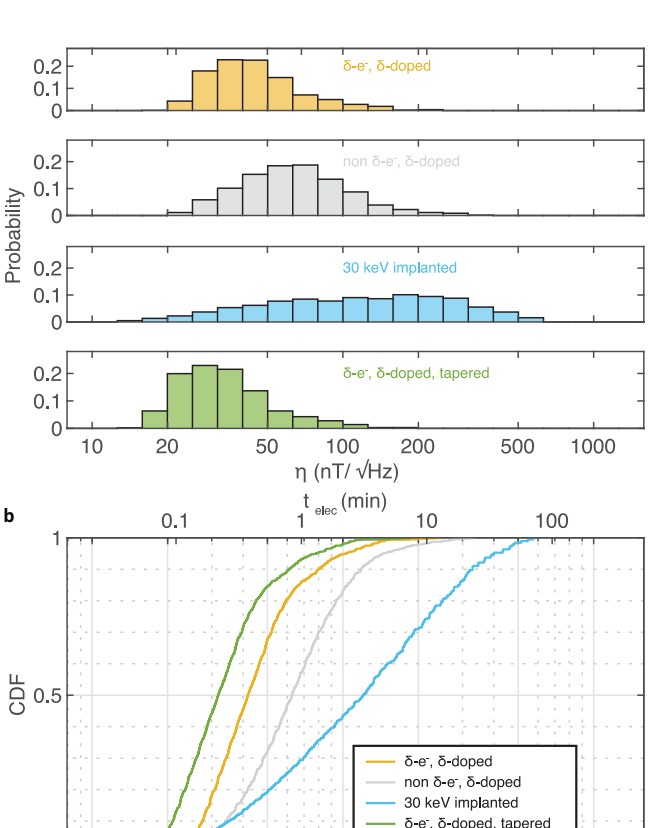

**Fig. 5 | High-yield fabrication of highly sensitive magnetic field sensors.**
**a** Histogram and **b** cumulative density function (CDF) of the estimated AC magnetic field sensitivity $\eta$ of single NVs in 480 nm pillars. The $\eta$ distribution is estimated for $\delta$-doped, $\delta$-electron irradiated (yellow) and $\delta$-doped, non-irradiated (gray) pillars using measured $PL_{sat}$, $T_2^{Hahn}$ and $C_{Rabi}$ distributions. The $\eta$ distribution of conventionally implanted pillars (cyan) is generated using $PL_{sat}$ and $C_{Rabi}$ measurements on our non-irradiated pillars with reported $T_2^{Hahn}$ distribution for 30 keV implantation[64]. The distribution for $\delta$-doped, $\delta$-electron irradiated pillar with better sidewall taper angle of 70° (green) is also estimated using $T_2^{Hahn}$ and $C_{Rabi}$ measurements on our $\delta$-electron irradiated pillars with the estimated PL improvement from FDTD simulations. A secondary $x$-axis shows the minimum averaging time for a 53 nm-deep NV to detect a single electron spin located on the diamond surface.

with long coherence and large $C_{Rabi}$ shown here is crucial for realizing advanced functionalities in devices for NV-based sensing, as discussed in the next section.

### High-yield, scalable magnetic field sensors
Lastly, we present an outlook for the improvements our method offers to scalable, high-yield fabrication of highly sensitive magnetic field sensors.

In a typical optical spin-state readout scheme, the alternating current (AC) magnetic field sensitivity $\eta$ is given as

$$\eta = \frac{\hbar}{g_e \mu_B} \frac{1}{e^{(-2\tau/T_2^{Hahn})}\sqrt{2\tau}} \sqrt{1 + \frac{4}{C_{Rabi}^2 n_{avg}}}, \qquad (3)$$

where $\hbar$ is the reduced Planck constant, $g_e \approx 2$ is the NV's electronic g factor, $\mu_B$ is the Bohr magneton, $2\tau$ is the total free evolution time, and $n_{avg}$ is the average photon number per measurement. This expression highlights the importance of long $T_2^{Hahn}$, large $C_{Rabi}$, and high PL for sensing small magnetic fields.

Figure 5a shows simulated histograms of $\eta$ for NVs in pillars formed via our method (yellow) compared to two other methods: conventional 30 keV nitrogen-implanted layers (cyan) and $\delta$-doped layers without lateral localization (gray). Also plotted are $\delta$-electron irradiated pillars with future improvements to the pillar geometry (green). The distributions are generated using Equation (3) with $n_{avg} = 0.5 \cdot PL_{sat} \cdot 400$ ns and $2\tau = T_2^{Hahn}$ with the $PL_{sat}$, $C_{Rabi}$ and $T_2^{Hahn}$ distributions experimentally measured in this work. From this histogram, we calculate the cumulative density function $P(X < \eta)$ for the corresponding sensitivity distribution $X$ in Fig. 5b. The median $\eta$ of NVs formed using our method is 42 nT/$\sqrt{\text{Hz}}$ with 86% of the NV centers exhibiting $\eta \lesssim 68$ nT/$\sqrt{\text{Hz}}$. For reference, a 53-nm deep NV with $\eta = 68$ nT/$\sqrt{\text{Hz}}$ can detect a single electron spin at the diamond surface in a typical averaging time of 1 minute[43]. For the non-localized $\delta$-doped method, the median $\eta$ is 63 nT/$\sqrt{\text{Hz}}$ with 57% of the NVs exhibiting $\eta \lesssim 68$ nT/$\sqrt{\text{Hz}}$, where we use the measured $PL_{sat}$, $C_{Rabi}$, and $T_2^{Hahn}$ distributions of non-irradiated pillars. With the conventional, implantation-based method, the median $\eta$ is 121 nT/$\sqrt{\text{Hz}}$ with 29% of the NVs exhibiting $\eta \lesssim 68$ nT/$\sqrt{\text{Hz}}$, where we use the measured $PL_{sat}$ and $C_{Rabi}$ distributions from non-irradiated pillars with the reported distribution of $T_2^{Hahn}$ from 30 keV implantation[64], chosen because it produces a similar NV depth of 40–50 nm. Hence, our method produces a significantly higher yield of high-sensitivity NV magnetometers, where we demonstrate an estimated 3-fold higher yield of single-electron-spin detectable magnetometers compared to conventional implantation-based methods. We note that further improvements can be realized by utilizing pillars with a 70° sidewall taper angle[34], as shown in green in Fig. 5. Higher-order dynamical decoupling can also extend coherence time, where an order of magnitude increase for shallow, $\delta$-doped NVs has been demonstrated[47], leading to a further 3× improvement in $\eta$ to <10 nT/$\sqrt{\text{Hz}}$. For comparison, the reported values of $\eta$ of single NV magnetometers for nanoscale imaging include 56 nT/$\sqrt{\text{Hz}}$[65], 56 nT/$\sqrt{\text{Hz}}$ (dynamical decoupling with 512-pulse XY8: 18 nT/$\sqrt{\text{Hz}}$)[66], 104 nT/$\sqrt{\text{Hz}}$[67], and 66 nT/$\sqrt{\text{Hz}}$ (dynamical decoupling with 128-pulse XY8: 32 nT/$\sqrt{\text{Hz}}$)[68].

## Discussion
To conclude, we demonstrate three-dimensionally localized formation of highly coherent NV centers aligned to prefabricated nanophotonic structures. Using our method, we find NV spin and photoluminescence properties superior to those for NV centers formed via conventional implantation methods as well as nonlocalized $\delta$-doped methods. These improved properties culminate in a significantly higher yield of high-sensitivity magnetometers, an important application of NV centers. Through our work, we also gain an understanding of vacancy diffusion in nanostructured diamond devices.

While we demonstrate our technique here on NV centers in diamond nanopillars, we emphasize that our method can be readily applied to other device geometries, such as 2D and 1D photonic crystal cavities[24,33] and nano-optomechanical devices[69] as well as to other material systems, including divacancies[70] and silicon vacancies[71] in silicon carbide and T centers in silicon[72]. In these optically addressable qubit systems, the advantages outlined here can be transferred using a similar targeted irradiation technique guided by our model of vacancy diffusion and capture for different geometries. The versatility of our method makes it attractive to apply to, for instance, NV center formation in (111)-oriented diamonds[42], with prospects of improved photon count rates in nanopillars[73], circular "bulls-eye" cavities for enhanced photon collection[74], as well as shallow (<-10 nm deep) NV centers, with applications in NV-driven magnetic resonance imaging and ultra-high spatial resolution imaging of condensed matter systems.

Looking forward, there is still room for further improvement in the collective control over the number, position, and coherence of color centers. The ultimate goal of forming a single defect with unit probability at a spot can be achieved by, for instance, $\delta$-electron irradiation

with in-situ annealing and photoluminescence characterization[51]. Lateral confinement can be enhanced by reducing the annealing time $t_{anneal}$, i.e., $\sigma_{loc} \propto t_{anneal}^{0.5}$, though the tradeoffs with potentially reduced coherence and NV yield will need to be optimized[45]. Reducing the nitrogen density within the $\delta$-doped layer is expected to extend $T_2^{Hahn}$ beyond 1 ms[46], and again, the tradeoff with NV number will need to be explored. Overall, our results strengthen the role of optically addressable solid-state spin defects in next-generation metrology and information science.

## Methods

### PECVD diamond growth
Diamond homoepitaxial growth and $\delta$-doping were performed via plasma-enhanced chemical vapor deposition (PECVD) using a SEKI SDS6300 reactor on a (100) oriented electronic grade diamond substrate (Element Six Ltd.). Prior to growth, the substrate was fine-polished by Syntek Ltd. to a surface roughness of ~200–300 pm, followed by a 4–5 μm etch to relieve polishing-induced strain. The growth conditions consisted of a 750 W plasma containing 0.1% $^{12}CH_4$ in 400 sccm $H_2$ flow held at 25 torr and ~730 °C according to a pyrometer. A ~154 nm-thick isotopically purified (99.998% $^{12}C$) epilayer was grown. During the nitrogen $\delta$-doping period of growth, $^{15}N_2$ gas 1.0% of the total gas content) is introduced into the chamber for five minutes. After growth, the sample was characterized with secondary ion mass spectrometry (SIMS) to estimate the isotopic purity, epilayer thickness, and properties of the $\delta$-doped layer (3.6 nm thick, 98 ppm nm, see Supplementary Section 2).

### NV characterization
All NV measurements are performed using a home-built confocal microscope using a 532 nm laser in an external magnetic field of 50–100 G. A 0.7 NA objective lens collects the emitted photons, which are then filtered 594 nm by a long-pass filter (SEMROCK BLP01-594R-25) and detected using an avalanche photodiode (SPCM-AQRH-14-FC). Microwave signals are delivered through an external gold wire.

### Monte Carlo simulations
We simulate NV center formation using MC simulation. Our simulation models the dominant effects during the annealing step, namely the diffusion of monovacancies within the prefabricated device and their capture by the existing nitrogen atoms to form NV centers. We also consider vacancy recombination at the diamond surfaces[39,75] (e.g., the top surface and the pillar's sidewalls for nanopillars).

Simulating the atomic-scale diffusion process on a diamond lattice is computationally intensive, necessitating approximately 50 million discrete "jumps" for a vacancy to traverse 1 μm, with each jump spanning a minuscule 0.154 nm C–C bond spacing. Therefore, we adopt a coarse-grained approach with a cubic lattice of spacing 1 nm, still significantly smaller than the device dimensions. Within each simulation run, we randomly select positions for $N_N$ nitrogen atoms within the $\delta$-doped layer region and $N_V$ vacancies within the vacancy-rich area from 200 keV electron irradiation. We estimate $N_V$ from CASINO simulations with a scaling factor $\alpha$, which is set as a free parameter (see Supplementary Section 4). For computational efficiency, we only consider vacancies with depth (<1 μm) since deeper vacancies do not contribute to NV center formation.

Given the initial conditions of the simulations, we segment the annealing process into shorter time steps. In each step, all vacancies randomly move some number of jumps, after which we check if they encountered a capture event. A capture by a nitrogen atom can occur when a monovacancy is in the same coarse-grained cell as a nitrogen atom with a probability of $\frac{16 V_{cc}/V_{uc}}{(8V_{cc}/V_{uc})^2/2}$, where $V_{cc}$ and $V_{uc}$ are the volumes of the coarse-grained cell and unit cell, respectively. When an NV center forms, both the vacancy and the nitrogen atom are removed from the simulation during the subsequent time steps. Conversely, if a monovacancy gets captured by the boundaries, only the vacancy is eliminated.

### Finite-difference time-domain simulations
Ansys Lumerical FDTD software is used to simulate the collection efficiency of photons emitted by NVs inside nanopillars. We model an NV as a point source consisting of two orthogonal dipoles perpendicular to the NV axis. The emission frequency range of the dipole is chosen to match the frequency range of the phonon sideband of the NV emission spectrum at room temperature: 650–800 nm. The NVs are positioned 53 nm below the top surface. We simulate our two nanopillar geometries with diameters 280 nm and 480 nm with the side wall angle of ~83° and height of 1.4 μm, which are attached to a diamond slab of finite thickness. For computational efficiency, we set the thickness of the slab to be 1 μm. To avoid any interference due to this relatively thin slab, we absorb all incoming fields at the bottom surface of the slab. We do this by setting the simulation area such that the bottom diamond interface matches the perfectly absorbing simulation boundary. The collection efficiency is then calculated from the power transmitted to a monitor plane just below the pillar inside the slab. We calculate the far-field emission through a collection cone with NA = 0.7.

To calculate the mean collection efficiency for the distribution of NVs with a given $\sigma_{loc}^{pillar}$, we first sweep the position of the NV laterally in two orthogonal directions ($dx$ and $dy$) and calculate the collection efficiency. Then, we extrapolate the collection efficiency for a given radial displacement $\vec{dr} = (dr\cos\theta, dr\sin\theta)$ by assuming the superposition of two orthogonal NVs. In particular, the collection efficiency is calculated as a weighted average of those calculated at two orthogonal displacements $dx = dr$ and $dy = dr$, where the weights are given as $\cos^2\theta$ and $\sin^2\theta$, respectively. We sweep the lateral confinement of NVs from perfectly localized ($\sigma_{loc}^{pillar} = 0$ nm) to maximally delocalized ($\sigma_{loc}^{pillar} = 1/4 \times$ pillar diameter, corresponding to a uniform lateral distribution across the pillar). For simplicity, we set the lateral probability distribution to follow a 2D Gaussian function with a spread of $\sigma_0$ which we truncate at the pillar boundary beyond which the probability is zero. For a given $\sigma_0$, we use the probability distribution and simulated collection efficiency to calculate both $\sigma_{loc}^{pillar}$ and the mean of the collection efficiency.

## Data availability
All the experimental data used in this work are available via Zenodo at https://doi.org/10.5281/zenodo.16747927.

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

## Acknowledgements

The authors thank Hitoshi Kato and JEOL Ltd. for 200 keV electron irradiation. We gratefully acknowledge the support of the Gordon and Betty Moore Foundation's EPiQS Initiative via Grant GBMF10279. We also acknowledge support from Cooperative Research on Quantum Technology (2022M3K4A1094777) through the National Research Foundation of Korea(NRF) funded by the Korean government (Ministry of Science and ICT(MSIT)), and from the DOE Q-NEXT Center (Grant No. DOE 1F-60579). The authors acknowledge the use of shared facilities of the UCSB Quantum Foundry through Q-AMASE-i program (NSF DMR1906325), the UCSB MRSEC (NSF DMR 1720256), the UCSB Nanofabrication Facility (an open access laboratory), and the Quantum Structures Facility within the UCSB California NanoSystems Institute. S.A.M acknowledges support from UCSB Quantum Foundry. L.B.H. acknowledges support from the NSF Graduate Research Fellowship Program (DGE 2139319) and the UCSB Quantum Foundry.

## Author contributions

S.K., P.L., D.Y., W.M., and A.B.J. conceptualized this work. S.K. and A.B.J. curated the data. S.K., P.L., and A.B.J. conducted the formal analysis and visualization of the results. S.K. and D.Y. performed the investigation on the diamond nanostructure fabrication and post-irradiation processing. L.H. also performed the investigation on PECVD growth of the δ-doped diamond sample. P.L. wrote the software code for Monte Carlo simulations. S.K., P.L., and D.Y. performed the confocal microscopy measurements on the final device. S.K., P.L., D.Y., L.H., W.M., and A.B.J. contributed to the methodology. A.B.J. was responsible for funding acquisition, project administration, resources, and supervision. S.K., P.L., D.Y., L.H., J.A., S.M., W.M., K.M., and A.B.J. contributed to the validation of the results and participated in the writing, review, and editing of the manuscript.

## Competing interests

The authors declare no competing interests.
