## [Transparent Peer Review file · Nature Communications]

Scalable, Nanoscale Positioning of Highly Coherent Color Centers in Prefabricated Diamond Nanostructures

Corresponding Author: Professor Ania Bleszynski Jayich

Version 0:

Reviewer comments:

Reviewer #1

(Remarks to the Author)

NV center in diamond is a leading solid-state spin quantum platform in quantum technology. Scalable, nanoscale positioning of highly coherent NV centers in nanostructures is useful for the applications in high-performance photonic devices and quantum technology. In this study, the authors use PECVD to incorporate a nitrogen δ -doped layer at a precisely controlled depth, followed by targeted 200 keV electron irradiation to generate localized vacancies. This method achieves depth confinement of approximately 4 nm and lateral positioning within tens of nanometers. The resulting NV centers exhibit consistently long spin coherence times ($T_2 \sim 98 \mu\text{s}$) and enhanced photoluminescence, leading to a markedly higher yield of high-sensitivity magnetometers. This approach represents a significant advancement in scalable high-performance NV center fabrication, so I recommend it for publication in Nature Communications. Below comments should be solved.

1. In this study, the authors utilize a 200 keV electron beam to generate vacancies. However, it remains unclear how variations in electron beam energy might influence lateral positioning accuracy. A discussion of energy-dependent effects on vacancy diffusion could help identify an optimal beam energy to further enhance NV center spatial precision.
2. In Figure 4e, the authors investigate the optical properties of NV centers in 280 nm and 480 nm pillars. They attribute the increase in PLsat for 480 nm pillars primarily to improved NV localization precision, while observing no significant dependence in 280 nm pillars. The authors are expected to clarify the underlying mechanism driving this difference. The non- δ -e data (such as Mean: 706(74) kcps) in Fig. 4c and 4d are not clear, please use a clear color.
3. The authors should compare the counts and coherence time of the NV centers with previous implanted methods (Sci. Adv. 8, eabn9573 (2022)).
4. The authors study the improved AC magnetic field sensitivity, how about the effect on DC magnetic field sensitivity?
5. The authors should add details of the laser confocal system, including the numerical aperture (NA) of the objective.
6. In the Supplementary information, Fig. S8 should be placed before the References.

Reviewer #2

(Remarks to the Author)

As we all know, diamond-based nanofabrication technology has been very mature since 10 years before. Specially, one can buy the commercial quantum scanning tips (See QST products in QZabre company) based on nitrogen-vacancy (NV) center in diamond always yield high spatial resolution (~ 10 nm) in quantum sensing field. In this article, the authors combined nitrogen δ -doping and δ -electron irradiation method to create NV centers in diamond with a high limitation on 3D spatial distribution. Maybe there is a little some things new in fabrication method. However, the properties of NV centers in this work is too poor and I strongly do not recommend their work for publication in the Nature Communications journal for current results. The detailed reasons are as follows:

1. The dark nuclear spins (e.g. Hydrogen in proteins which is out of diamond) detection is the theme for NV center based nano-NMR technology. In this topic, the depth (distance between diamond surface and NV center) of NV center is always about 5 nm or less (See Xu, Zhewen, et al. Minimizing sensor-sample distances in scanning nitrogen-vacancy magnetometry. ACS Nano 19 (8), 8255-8265 (2025)). Hence, the NV centers, which are created in this job, is too deep too touch this topic.

2. Once the depth of NV center in diamond is large than 20 nm, the spin coherence of NV center will be very long. For example, the group of Maletinsky, P. demonstrated the fabrication of single-crystalline diamond nanopillars and NV center created with traditional method with an average T_2 time of $235 \pm 46 \mu\text{s}$, 25% signal contrast under continuous laser excitation and 1 Mcps Photon count rate in 2014 (See: Neu, Elke, et al. Photonic nano-structures on (111)-oriented diamond. Appl. Phys. Lett. 104, 153108 (2014)). So the spin coherence and contrast of NV centers in this work are much poor than results from more than 10 years ago, which makes this work mediocre.

3. The main results on the spatial distribution of NV centers need to be measured using different methods to ensure their correctness (See Pham, Linh M., et al. NMR technique for determining the depth of shallow nitrogen-vacancy centers in diamond. Phys. Rev. B 93, 045425 (2016) or Wang, Junfeng, et al. Coherence times of precise depth controlled NV centers in diamond. Nanoscale 8, 5780-5785 (2016)).

4. The second order autocorrelation function of NV center, which can be used to count the number of NV center in experiment, should be measured and provided in this work.

5. Since the number of fluorescent photons is increased with nanofabrication technology, the SNR of signal detection is improved and the number of NV center under investigation should be increase up to ~ 100 to decrease the significant fluctuation in Fig.4 in the main text.

Reviewer #3

(Remarks to the Author)

Overall comments:

I'd like to commend the authors for putting together a very thorough set of experiments supported by carefully done simulations. Building upon several well established techniques, this is an elegant step forward for enhancing precision placement of NV centers in diamond nanopillars without sacrificing the performance of the NV.

Overall the work is well laid out, and my comments for improvement are mostly based on linking the significance of the work to the larger field, particularly referencing specific works when comparing results. Additionally, the work mentions the relevance to NV magnetometers. Including more comments and references on how the work compares to the state of the art would increase the significance of the work and the reach of the reader base. More specific comments are given below.

Providing more clarity on the link to sensing applications:

The authors do an excellent job of taking the reader step by step through the fabrication process to the measurements and significance. However, the relevance of the work to sensing applications is not highlighted well in the abstract, introduction or conclusion. The connection to improved sensing applications is explained well at the end of the "Spin and optical properties of single NVs in nanopillars section", particularly lines 275 through 278. Adding this kind of statement to the abstract, introduction and conclusion would help guide the reader more clearly and increase the reach of the work from fabrication focused readers to also sensing focused readers.

The same may be said of the "High-yield, scalable magnetic field sensors" section. The section does an excellent job laying out how the sensitivity of the nanopillars may be calculated with the improved methods, but does not connect well to the larger idea of the manuscript.

A note on missing citations/putting the results into more context:

Lines [301]-[304] mention that the sensitivity of a 53nm deep NV is $68\text{nT}/\sqrt{\text{Hz}}$, but do not reference any specific works. Please include references here.

Additionally, lines [317] and [318] mention that with further work, the sensitivity can be improved to $<10\text{nT}/\sqrt{\text{Hz}}$, however, there is no reference given for how this compares to the state of the art for NV magnetometers. Please include a note on how the overall sensitivity compares with the state of the art sensitivity limits for NV-based magnetometers with references to provide context for the reader.

Lines [209-210] cite reference [41] as the imaging technique used, but it is too vague. Please add in a half or one sentence to just explain the basic technique used so this piece can become more standalone.

Lines [214-215] state the majority of the σ_{loc} can be attributed to the annealing process, which may be apparent to a reader who is experienced in the field, but adding in a statement to explain why this is the case to the non-experienced

reader would add clarity here.

Line [265] mentions the improvement factors to the mean PLSat, but does not provide the number for PLSat. Please provide the number for the mean PLSat to provide context to the reader.

Line [271-272] mentions that the the PLSat does not depend strongly on the 280 nm pillars. Is this simply because the pillar is smaller than the 480 nm? Please provide one sentence to explain this further.

The conclusion:

The authors very well state in the conclusion that their fabrication techniques may be used for other geometries of diamond. Stating this more clearly in the abstract and introduction would help increase the significance of the work.

Figures:

- For all figures, please include the number for the scale bar on the image itself (i.e. 20um, 1um).
- The figures do not adapt very well to greyscale. For accessibility purposes, please increase the contrast of the colour schemes (i.e. purple and red in greyscale are not distinguishable). Additionally, some of the lighter colours (particularly in figure 3, are difficult to see in greyscale.)

References:

Reference [7] has been published. Please replace the arXiv reference with the updated link <https://www.science.org/doi/10.1126/sciadv.adp6442>

It would not appear reference [42] has yet been published, but please review again if possible before submission and update the citation if published.

Though I am aware it is not expected of reviewers to comment on grammatical and technical writing errors, I did collect a number of very small errors that may help act as a fresh pair of eyes for the authors to correct. I have outlined them below, referring to the line on the manuscript:

[039]: "we report positioning precision..." This is awkwardly worded

In a number of places, the citation is placed after a word without a space between the last word and the reference number. Please add a space to make consistent throughout the text. I've referenced all the ones I could see below: [050,051,053,055,071,072,078,081,085,086,090,092,094,095,148,169,248,310,315,335,336]

The sentence in lines [101-104] is somewhat unclear. Perhaps structuring the sentence inversely would make it more clear to the reader.

[352]: hyphen missing in delta doping (delta-doping)

[363]: Possible typo in the unit of 98 ppm*nm ?

Reviewer #4

(Remarks to the Author)

The manuscript reports a controlled creation of NV color centers in prefabricated diamond nanopillars and characterization of their properties, primarily regarding nanoscale applications such as quantum sensing. It is a nice work built upon the expertise of the authors in diamond wafer growth, color center formation, nanofabrication, and quantum sensing. I would like to raise following questions and make some comments:

Main questions:

Q1. I am not sure if the 'estimated (not measured with a test field)' AC sensitivity of 42 nT/sqrt(Hz) (see fig5 and associated discussion) is a significant breakthrough. Because I would like to cite a few earlier works and want to understand what is (are) the main factor(s) limiting the sensitivity here. First, if I am not wrong, the NV centers in the studied sample here are surrounded by an isotopically pure (13-C depleted) carbon environment, and the previous works (e.g., Nature Materials, volume 8, page 383 (2009)) on the isotopically pure samples showed AC (Hahn) sensitivity of 4 nT/sqrt (Hz) per single 'bulk' NV. Second, in a non-isotopic sample, Grinolds et al (Nature Physics volume 9, page 215 (2013)), achieved 56 nT/sqrt(Hz)

AC (Hahn) sensitivity per single NV in a nanopillar.

So, I want to know despite having a nanopillar and a pure sample, why the sensitivity has not improved substantially?

Q2. Comparing with the work of Wang et al., Sci. Adv. 8, eabn9573 (2022) which the authors cite in this manuscript (reference 34), I think the results in the manuscript show significant enhancement of T₂ (Hahn) while relatively lower precision in NV localization. So, with this trade off, can the authors highlight/envision what specific advantage their method achieves in quantum sensing over the performance (in quantum sensing) of devices reported in the ref. 34?

Q3. Is not the depth (~50 nm) of NV centers in this case is too deep for sensing nuclear spins on the diamond surface for doing molecular NMR/MRI? Also, is the predicted median AC sensitivity of ~40 nT/sqrt (Hz) is sufficient to sense nuclear spins external to the diamond surface?

For e.g., if the field produced by an electron situated at 50 nm away (diamond surface) is ~50 nT then the field produced by a single proton at similar distance will be in tens of pT, which requires far more sensitivity than the that is achieved (even with the dynamical decoupling sequences applied to NV centers studied here) in this work. So, in which area of quantum sensing/specific applications would their devices be advantageous? Can the authors make any such remarks about it in the discussion part?

Q4. Fig3c: There is a huge error bar for the smaller doses (esp. those close to 10¹⁹ e-/cm²) and the errors systematically reduce at higher doses. I assume this is simply because of the less probability of creating vacancies (and hence the NV complexes during annealing) for the low irradiation dose. Can the authors clarify this point and explain by adding more information in the text to help the reader?

Q5. The spatial confinement (NV positioning precision in nanopillars) they are arriving at is based on the simulations since apparently the direct (experimental) measurements of this uncertainty is challenging due to influence of pillars' photonic modes. How reliable (in approximate percentage terms) is the method compared to measuring the positioning precision directly, say experimentally?

Other questions:

Q1. Given the deleterious effect of nanostructuring on the optical coherence/spectral properties of the NV centers, does this method also produce optically coherent NV centers for quantum networking applications? I know optimizing the optical coherence is probably not the main goal of this paper, but authors do seem to suggest their devices have generic quantum applications beyond the sensing.

Q2. The authors show 1.8 times enhancement of PL compared to non-irradiated pillars. Can the author comment on how does their absolute count rate compare (by taking into account the NA of their collection microscope objective) with the nanopillars created by other works for similar NV depth below the surface?

Q3. Ref. 34 reports about 30% Rabi contrast while here the average Rabi contrast is 18%. What is the limiting factor here? Is it possible to further boost the Rabi contrast in these devices to 30%?

Q4. Fig1 c inset: what is the 532 nm laser power for obtaining the PL confocal scans of the nanopillars? Please include it in the caption at least.

Q5. Page4: What is the need for doing air annealing at 450C, given that the NVs are fairly deep (>50 nm) below the surface? If this surface termination is not done, does the T₂ get worse? If so, by what amount?

Q6. In the concluding part, the authors allude to improving the coherence in the future works but do not mention what specific potential strategies could be employed to increase the coherence (T₂) beyond their current 100 us range.

Q7. Also in the concluding part, I feel the authors' suggestions for gaining more control over number and the positioning accuracy of the defects in nanopillars is a bit vague. For example, how does one achieve "in-situ annealing"? Similarly, the authors state in the same paragraph, "More accurate positioning can be achieved by reducing the annealing time"—reduction by what amount and at the cost of what (NV creation yield, coherence etc.)?

Some of it is already mentioned in the main text while discussing the results of fig3, it would also help the reader if little more details are added to the concluding section with some appropriate literature references.

Q8. How do authors verify that the NV centers they are studying are singles? Have they performed the standard antibunching measurements to prove this?

Q9. Page7, line 298-299, why is the multiplicative factor of 400 ns used to get the average number of photons?

Some comments:

C1. Page6 and fig4a: Authors report a measured T₂ value of nearly 100 us from the nanopillars but do not clarify whether it is from the 280 nm diameter pillar or from the 480 nm nanopillar.

C2. Potentially misleading: The statement which seems to be highlighted in the abstract is that the lateral localization

precision is 46 nm. This number for precision is true but corresponds to 280 nm diameter pillars. However, one of the supposedly key findings of the paper, the PL saturation counts (PL_sat; 1.8X) is quoted from the 480 nm diameter pillar for which the NV localization precision is actually 72 nm. Similarly, I think it is not clear whether the T2 value of ~100 us is measured on pillars with dia. 280 nm or with dia. 480 nm. Finally, the best value for sensitivity (fig5) was also estimated based on the 480 nm pillar. If 480 nm pillars are their best devices (in terms of achievable sensitivity) then I would state the localization precision corresponding to that pillar (72 nm, not 46 nm). Anyway, I can understand that the authors might be intended to quote the best values from their work, however, in that case it would be fair to explicitly quote the pillar diameter for the key findings (localization precision, PL_sat, and T2) in the abstract (and wherever possible) to avoid the ambiguity for the reader.

C3. It would be better to clarify in the main text that the saturation counts shown in the fig4 are already accounted for the counts from the background.

C4. Page5, first line, text in the bracket. I think the confocal of unpatterned mesas are in 'top right and bottom left' of fig2 while their positions are rotated in fig1. Accordingly the text (in page5) needs to be corrected. Please maintain the constituency.

C5. Wherever the supplementary material is quoted, it would be better to specifically mention which section and figures of the supplementary material so that the reader is appropriately directed.

Version 1:

Reviewer comments:

Reviewer #1

(Remarks to the Author)

The authors have addressed my concerns and improved the manuscript, so I recommend it for publication in Nature Communications.

Reviewer #2

(Remarks to the Author)

I have read the revised manuscript and responses to reviewers. The novelty of this study lies in combining nitrogen δ -doping and δ -electron irradiation to create NV centers in diamond with a high limitation on 3D spatial distribution. The created NV centers have some comparable properties, which can be applied in multiple scenarios. The revised manuscript has addressed most of the comments and improved its quality.

Reviewer #3

(Remarks to the Author)

Thank you to the authors for this second version of the manuscript.

The manuscript has been well laid out, taking into account the recommendations by reviewers and follows a nice rhetoric making the results and impact clear. The relation to applications on this second version is clear and the reader can clearly understand where the methods used in this paper may be applied for further NV based applications.

I have no further follow-up points for corrections.

Reviewer #4

(Remarks to the Author)

I am pleased to see the revised version of the manuscript, which satisfactorily addresses all of my previous questions, concerns, and comments. The authors have provided clear, precise, and transparent responses, and have made substantial additions and corrections to both the main text and the supplementary materials. These changes significantly enhance the overall scientific quality and clarity of the work.

One of my principal concerns was understanding how this study differentiates itself from recent reports by other groups employing similar nanophotonic platforms. In this regard, the authors have provided a convincing and well-argued justification that clearly establishes the novelty and relevance of their contribution within the existing literature.

In light of these improvements, I support the publication of this manuscript in Nature Communications.

Minor Note (trivial): In one of my earlier comments (C4), I mistakenly wrote "Please maintain the constituency"; the intended word was "consistency."

Response to Referees on “Scalable, nanoscale positioning of highly coherent color centers in prefabricated diamond nanostructures”

We are very grateful to all the reviewers for their detailed reading of our manuscript and their helpful suggestions. Below, we respond point by point to the reviewer comments, and we indicate where in the main text and supplement we have made changes. In responding to the comments, we find our manuscript greatly strengthened.

Reviewer 1

NV center in diamond is a leading solid-state spin quantum platform in quantum technology. Scalable, nanoscale positioning of highly coherent NV centers in nanostructures is useful for the applications in high-performance photonic devices and quantum technology. In this study, the authors use PECVD to incorporate a nitrogen δ -doped layer at a precisely controlled depth, followed by targeted 200 keV electron irradiation to generate localized vacancies. This method achieves depth confinement of approximately 4 nm and lateral positioning within tens of nanometers. The resulting NV centers exhibit consistently long spin coherence times ($T_2 \sim 98 \mu\text{s}$) and enhanced photoluminescence, leading to a markedly higher yield of high-sensitivity magnetometers. This approach represents a significant advancement in scalable high-performance NV center fabrication, so I recommend it for publication in Nature Communications. Below comments should be solved.

1. In this study, the authors utilize a 200 keV electron beam to generate vacancies. However, it remains unclear how variations in electron beam energy might influence lateral positioning accuracy. A discussion of energy-dependent effects on vacancy diffusion could help identify an optimal beam energy to further enhance NV center spatial precision.

We thank the reviewer for their careful reading of our manuscript and their helpful questions and suggestions. Regarding this first question, we do not expect the lateral positioning accuracy to be strongly affected by the beam energy. Simulations shown in Fig. 1 below indicate that the lateral vacancy distribution after irradiation does not depend significantly on electron energy and rather, it is set by the spot size of the electron beam, which was 20 nm in this work. The annealing after irradiation plays the largest role in the final lateral positioning accuracy. Nevertheless, in the future, we do intend to vary the electron energy, and will be able to experimentally test these assertions.

Fig. 1 CASINO simulation of vacancy creation. Vacancy creation sites from 1×10^6 electron trajectories for different irradiation energies.

Following the reviewer’s comment, we have included a paragraph and a new figure (same as Fig. 1 of this response) in the Supplementary Information section ‘Vacancy formation via δ -electron irradiation’: **“In Fig. S3(e), we also investigate monovacancy creation from different irradiation energies. ...”**

2. In Figure 4e, the authors investigate the optical properties of NV centers in 280 nm and 480 nm pillars. They attribute the increase in PLsat for 480 nm pillars primarily to improved NV localization precision, while observing no significant dependence in 280 nm pillars. The authors are expected to clarify the underlying mechanism driving this difference. The non- δ -e data (such as Mean: 706(74) keps) in Fig. 4c and 4d are not clear, please use a clear color.

We thank the reviewer for the comment regarding the photon counts for 280 nm and 480 nm pillars. Following the reviewer’s comment, we have added the following sentences to the main text subsection “Spin and optical properties of single NVs in pillars”: **“This can be attributed to a higher-order waveguiding mode, TE01, strongly outcoupling photons from NVs close to the edge of 280 nm pillars. In contrast, for 480 nm pillars, NVs close to the edge are weakly coupled to the TE01 mode, while NVs close to the center are coupled to both the fundamental mode HE11 and the higher order mode HE12, thus exhibiting sharper PL dependence on NV lateral displacement.”**

Also, we have adjusted the color for the non- δ -e irradiated data to be darker and hence more clear.

3. The authors should compare the counts and coherence time of the NV centers with previous implanted methods (Sci. Adv. 8, eabn9573 (2022)).

We have included the following text in the section on NV optical properties: **“Higher PL_{sat} values could be achieved with pillar geometries optimized for increased collection efficiency, such as parabolic pillars [*Physical Review Applied* 14, 064007 (2020)] or tapered pillars [*Science Advances* 8, eabn9573 (2022)], where with pillars with a $\sim 70^\circ$ sidewall angle, the authors achieved an average $PL_{sat} = 1.6$ Mcps.”** We also note that we did state the coherence times measured in (Sci. Adv. 8, eabn9573 (2022)) in both the Introduction section and the “Spin and optical properties of single NVs in nanopillars” section.

4. The authors study the improved AC magnetic field sensitivity, how about the effect on DC magnetic field sensitivity?

We expect that η_{DC} (DC magnetic field sensitivity) would be improved as well compared to non- δ -irradiated pillars, because η_{DC} also depends on PL_{sat} and Rabi contrast. However, for the NVs presented here, we did not perform a systematic measurement of T_2^* , a relevant parameter needed for estimating η_{DC} .

5. The authors should add details of the laser confocal system, including the numerical aperture (NA) of the objective.

A subsection about the confocal system, i.e., NV characterization, has been to the Methods section.

6. In the Supplementary information, Fig. S8 should be placed before the References.

We have modified the Supplementary Information according to the reviewer’s comment.

Reviewer 2

As we all know, diamond-based nanofabrication technology has been very mature since 10 years before. Specially, one can buy the commercial quantum scanning tips (See QST products in QZabre company) based on nitrogen-vacancy (NV) center in diamond always yield high spatial resolution (10 nm) in quantum sensing field. In this article, the authors combined nitrogen δ -doping and δ -electron irradiation method to creat NV centers in diamond with a high limitation on 3D spatial distribution. Maybe there is a little some things new in fabrication method. However, the properties of NV centers in this work is too poor and I strongly do not recommend their work for publication in the Nature Communications journal for current results. The detailed reasons are as follows:

1. The dark nuclear spins (e.g. Hydrogen in proteins which is out of diamond) detection is the theme for NV center based nano-NMR technology. In this topic, the depth (distance between diamond surface and NV center) of NV center is always about 5 nm or less (See Xu, Zhewen, et al. Minimizing sensor-sample distances in scanning nitrogen-vacancy magnetometry. *ACS Nano* 19 (8), 8255-8265 (2025)). Hence, the NV centers, which are created in this job, is too deep too touch this topic.

We thank the reviewer for their evaluation of our manuscript. We would like to make two points in response to this first comment:

1) Though NV center-based nano-MRI is indeed an exciting frontier, there are many other NV-based sensing applications that do not demand NV-target separations of < 5 nm. For example, NV centers located 20-100 nm below the diamond surface have been used to elucidate various nanomagnetic and nanoelectronic phenomena in a variety of condensed matter systems. Further, in scanning-NV experiments that utilize diamond scanning probes for probing, *e.g.* magnetic materials, NV-sample distances of < 10 nm are in practice rarely achieved due to the difficulty of minimizing sensor-sample distance. This discrepancy between NV depth and NV-sample separation is well-known and discussed in the NV-scanning probe microscopy (NV-SPM) community and has recently been reported on - see Table 1 of ACS Nano 19 (8), 8255-8265 (2025)). Hence, for NV-SPM applied to condensed matter systems, a 50 nm NV depth is often more than sufficient, and further yields significantly more stable NV centers that retain their properties for long measurement times.

2) Our method is completely compatible with the shallow (< 10 -nm-deep) NV centers that are desired for nanoMRI and we look forward to exploring δ -electron irradiated or δ -doped shallow NV centers in the future (or for others to apply our technique to shallow δ -doped NV centers). In response to the reviewer's question regarding applicability to nanoMRI, we have added the following sentence in the Conclusions: **"The versatility of our method makes it attractive to apply to, for instance, ... shallow ($< \sim 10$ nm deep) NV centers, with applications in NV-driven magnetic resonance imaging and ultra-high spatial resolution imaging of condensed matter systems."**

2. Once the depth of NV center in diamond is large than 20 nm, the spin coherence of NV center will be very long. For example, the group of Maletinsky, P. demonstrated the fabrication of single-crystalline diamond nanopillars and NV center created with traditional method with an average T_2 time of $235 \pm 46 \mu s$, 25% signal contrast under continuous laser excitation and 1 Mcps Photon count rate in 2014 (See: Neu, Elke, et al. Photonic nano-structures on (111)-oriented diamond. Appl. Phys. Lett. 104, 153108 (2014)). So the spin coherence and contrast of NV centers in this work are much poor than results from more than 10 years ago, which makes this work mediocre.

Indeed, for NV centers deeper than 20-25 nm, NV properties can approach those of the bulk. We note, importantly, however, that the long T_2 NVs observed in the paper referenced by the reviewer (Neu et al, APL 104, 153108 (2014)) were 'native' NV centers with no depth control and hence are not useful for the reliable formation of NV-magnetometers. To achieve depth-controlled NV centers, either implantation or δ -doping should be pursued. Implantation leads to significantly more scatter in T_2 times than δ -doped NV centers. For instance, 15 keV ion implantation results in NVs with the majority (> 90 %) having T_2 less than $50 \mu s$ [reference 34 of our work] and for 30 keV ion implantation, corresponding to 40 nm NV depth, Figure 3d of Jakobi et al (doi:10.1088/1742-6596/752/1/012001) shows significant scatter of T_2 times. An important advantage of our technique is the reliably long T_2 time achieved, allowing for more scalable and reliable production of, *e.g.*, NV-based sensors.

We also note that the (111) diamond orientation in Neu et al, APL 104, 153108 (2014) increases photon count rates (due to optimal coupling of NV emission to the nanopillar mode), and combined with the deeper depths results in improved contrast in the (111)-oriented pillars. However, (111)-diamond and (111)-diamond pillars are still rarely used today (for many reasons, the main one being the difficulty of procuring high quality, low-nitrogen substrates). Hence, it is important to compare our results to (100)-oriented diamonds.

Lastly, we again note that our method is compatible with (111)-oriented diamonds, and it will be interesting to pursue this direction in the future, potentially leveraging recent developments in δ -doping of (111)-oriented diamond [1]. We have added the following sentence in the Conclusions: **“The versatility of our method makes it attractive to apply to, for instance, NV center formation in (111)-oriented diamonds [Physical Review X 15, 021035 (2025)] with prospects of improved photon count rates in nanopillars [Applied Physics Letters 104, 153108 (2014)],...”**

3. The main results on the spatial distribution of NV centers need to be measured using different methods to ensure their correctness (See Pham, Linh M., et al. NMR technique for determining the depth of shallow nitrogen-vacancy centers in diamond. Phys. Rev. B 93, 045425 (2016) or Wang, Junfeng, et al. Coherence times of precise depth controlled NV centers in diamond. Nanoscale 8, 5780-5785 (2016)).

In this work, we measure the spatial *depth* distribution of ^{15}N via secondary ion mass spectrometry (see Fig. S2), which shows a 53-nm-deep nitrogen layer confined to 3.66 nm. This method is very accurate when, as is the case in our samples, the N density is high enough to be detected by SIMS. The NVs are too deep to use the method of Pham et al. The ^{15}NVs , which we confirm from hyperfine-resolved ESR spectroscopy, are derived from this δ -doped layer. The majority of our work is focused on characterizing the *lateral* spatial distribution afforded by our technique, with the results presented in the section ‘Targeted formation of NVs in nanostructures’ and Fig. 3.

4. The second order autocorrelation function of NV center, which can be used to count the number of NV center in experiment, should be measured and provided in this work.

We did perform $g^{(2)}$ autocorrelation measurements to help identify single NVs that were used in Figs. 4 and 5 and the results have been added to the supplement and the following text has been added to the main text: **“, confirmed by second-order correlation measurements (see Supplementary Section 8) and hyperfine-resolved pulsed ESR measurements.”**

5. Since the number of fluorescent photons is increased with nanofabrication technology, the SNR of signal detection is improved and the number of NV center under investigation should be increase up to 100 to decrease the significant fluctuation in Fig.4 in the main text.

Indeed, more data would decrease the fluctuations in Fig. 4. However, taking a $g^{(2)}$ measurement on ~ 300 more pillars, and then measuring PL_{sat} , T_2 , and contrast on all the single NV pillars would be quite time consuming. We believe that the error bars we have achieved with the approximately 25 measurements are sufficient to support our claims.

Reviewer 3

Overall comments: I'd like to commend the authors for putting together a very thorough set of experiments supported by carefully done simulations. Building upon several well established techniques, this is an elegant step forward for enhancing precision placement of NV centers in diamond nanopillars without sacrificing the performance of the NV.

Overall the work is well laid out, and my comments for improvement are mostly based on linking the significance of the work to the larger field, particularly referencing specific works when comparing results. Additionally, the work mentions the relevance to NV magnetometers. Including more comments and references on how the work compares to the state of the art would increase the significance of the work and the reach of the reader base. More specific comments are given below.

We would like to first thank the reviewer for their evaluation of our manuscript and their helpful comments, in particular making the relevance of our work to wider applications and state-of-the-art more clear.

For a comparison of PL_{sat} to other work, we now write: **“Higher PL_{sat} values could be achieved with pillar geometries optimized for increased collection efficiency, such as parabolic pillars [*Physical Review Applied* 14, 064007 (2020)] or tapered pillars [*Science Advances* 8, eabn9573 (2022)], where with pillars with a $\sim 70^\circ$ sidewall angle, the authors achieved $PL_{sat} = 1.6$ Mcps.”**

For sensitivity comparisons, we have added: **“For comparison, the reported values of η of single NV magnetometers for nanoscale imaging include 56 nT/ $\sqrt{\text{Hz}}$ [*Nature Nanotechnology* 7, 320–324 (2012)], 56 nT/ $\sqrt{\text{Hz}}$ (dynamical decoupling with 512-pulse XY8: 18 nT/ $\sqrt{\text{Hz}}$) [*Nature Physics* 9, 215–219 (2013)], 104 nT/ $\sqrt{\text{Hz}}$ [*Nature Physics* 17, 1216–1220 (2021)], and 66 nT/ $\sqrt{\text{Hz}}$ (dynamical decoupling with 128-pulse XY8: 32 nT/ $\sqrt{\text{Hz}}$) [*Physical Review Applied* 17, 054008 (2022)].”**

Providing more clarity on the link to sensing applications: The authors do an excellent job of taking the reader step by step through the fabrication process to the measurements and significance. However, the relevance of the work to sensing applications is not highlighted well in the abstract, introduction or conclusion. The connection to improved sensing applications is explained well at the end of the “Spin and optical properties of single NVs in nanopillars section”, particularly lines 275 through 278. Adding this kind of statement to the abstract, introduction and conclusion would help guide the reader more clearly and increase the reach of the work from fabrication focused readers to also sensing focused readers.

The same may be said of the “High-yield, scalable magnetic field sensors” section. The section does an excellent job laying out how the sensitivity of the nanopillars may be calculated with the improved methods, but does not connect well to the larger idea of the manuscript.

Following the reviewer’s suggestion, we have made the following changes. In the Abstract, we write: **“Our method can improve the performance of various NV-based devices. In the realm of magnetic sensing, we achieve a $3\times$ improved yield**

of NV centers with single electron-spin sensitivity over conventional implantation-based methods”. In the Conclusion, we write: **“The versatility of our method makes it attractive to apply to, for instance, NV center formation in (111)-oriented diamonds [Physical Review X 15, 021035 (2025)] with prospects of improved photon count rates in nanopillars [Applied Physics Letters 104, 153108 (2014)], circular “bull’s-eye” cavities for enhanced photon collection [Nano Letters 15, 1493–1497 (2015)], as well as shallow (<~ 10 nm deep) NV centers, with applications in NV-driven magnetic resonance imaging and ultra-high spatial resolution imaging of condensed matter systems.”**

A note on missing citations/putting the results into more context: Lines [301]-[304] mention that the sensitivity of a 53nm deep NV is 68nT/sqrt(Hz), but do not reference any specific works. Please include references here.

The following reference is added: *Applied Physics Letter* 101, 082413 (2012).

Additionally, lines [317] and [318] mention that with further work, the sensitivity can be improved to <10nT/sqrt(Hz), however, there is no reference given for how this compares to the state of the art for NV magnetometers. Please include a note on how the overall sensitivity compares with the state of the art sensitivity limits for NV-based magnetometers with references to provide context for the reader.

We refer the reviewer to our response to their first comment, where we have added text comparing our sensors to state-of-the-art sensors for NV-based nanoscale magnetic imaging. The following sentences are added in the third subsection, i.e., High-yield, scalable magnetic field sensors, of the Main section: **“For comparison, the reported values of η of single NV magnetometers for nanoscale imaging include 56 nT/ $\sqrt{\text{Hz}}$ [*Nature Nanotechnology* 7, 320–324 (2012)], 56 nT/ $\sqrt{\text{Hz}}$ (dynamical decoupling with 512-pulse XY8: 18 nT/ $\sqrt{\text{Hz}}$) [*Nature Physics* 9, 215–219 (2013)], 104 nT/ $\sqrt{\text{Hz}}$ [*Nature Physics* 17, 1216–1220 (2021)], and 66 nT/ $\sqrt{\text{Hz}}$ (dynamical decoupling with 128-pulse XY8: 32 nT/ $\sqrt{\text{Hz}}$) [*Physical Review Applied* 17, 054008 (2022)].”**

Lines [209-210] cite reference [41] as the imaging technique used, but it is too vague. Please add in a half or one sentence to just explain the basic technique used so this piece can become more standalone.

The following text has been added to the first subsection, i.e., Targeted formation of NVs in nanostructures, of the Main text: **“... and individually fitting them also to a 2D Gaussian function $PL_{single}(x, y) \propto e^{-(x^2+y^2)/(2\sigma_{PSF}^2)}$ (see Supplementary Section 6)”** —

Lines [214-215] state the majority of the σ_{loc} can be attributed to the annealing process, which may be apparent to a reader who is experienced in the field, but adding in a statement to explain why this is the case to the non-experienced reader would add clarity here.

We have added the following to the first subsection (Targeted formation of NVs in nanostructures) of the main text, : **“Since the monovacancies created by δ -electron irradiation are highly localized (see Figure 1 and Supplementary Section 3),”** —

Line [265] mentions the improvement factors to the mean PLsat, but does not provide the number for PLsat. Please provide the number for the mean PLsat to provide context to the reader.

The values of PL_{sat} for the non-irradiated pillars (0.706(74) and 0.585(189) Mcps for 280 nm and 480 nm pillars, respectively) are displayed next to the data in Fig. 4c and 4d. Following a comment from Reviewer 1, we have adjusted the color so as to make these values more readable.

Line [271-272] mentions that the the PLsat does not depend strongly on the 280 nm pillars. Is this simply because the pillar is smaller than the 480 nm? Please provide one sentence to explain this further.

We thank the reviewer for the comment regarding the photon counts for 280 nm and 480 nm pillars. Following the reviewer’s comment, we have added the following sentences to the main text subsection “Spin and optical properties of single NVs in pillars”: **“This can be attributed to a higher-order waveguiding mode, TE01, strongly outcoupling photons from NVs close to the edge of 280 nm pillars. In contrast, for 480 nm pillars, NVs close to the edge are weakly coupled to the TE01 mode, while NVs close to the center are coupled to both the fundamental mode HE11 and the higher order mode HE12, thus exhibiting sharper PL dependence on NV lateral displacement.”**

The conclusion: The authors very well state in the conclusion that their fabrication techniques may be used for other geometries of diamond. Stating this more clearly in the abstract and introduction would help increase the significance of the work.

We address this alongside the first comment and reiterate our response for the reviewer’s convenience.

Indeed, our method is not limited to nanopillars; it is an example for showcasing the ability of our method. Following the reviewer’s suggestion, we have added the following snippets of text to the Abstract and Introduction: **“ Our method can improve the performance of various NV-based devices.”**, **“... with applications in, e.g., nanoscale magnetic sensing and photon-driven entanglement of solid-state emitters.”**, **“Our method is readily applicable to the formation of improved, emitter-containing devices in a variety of application spaces as well as device geometries beyond the nanopillars used here.”** .

—
Figures: - For all figures, please include the number for the scale bar on the image itself (i.e. 20um, 1um). - The figures do not adapt very well to greyscale. For accessibility purposes, please increase the contrast some of the colour schemes (i.e. purple and red in greyscale are not distinguishable). Additionally, some of the lighter colours (particularly in figure 3, are difficult to see in greyscale.)

We have labeled all the scale bars on the images. We have also increased the contrast between purple and red in Figures 2 and 3. Lighter colors in Figure 3 have been made darker. We have verified that all colors are distinguishable in grayscale.

References: Reference [7] has been published. Please replace the arXiv reference with the updated link <https://www.science.org/doi/10.1126/sciadv.adp6442>

It would not appear reference [42] has yet been published, but please review again if possible before submission and update the citation if published.

Both of the references have been updated to the published version.

—
Though I am aware it is not expected of reviewers to comment on grammatical and technical writing errors, I did collect a number of very small errors that may help act as a fresh pair of eyes for the authors to correct. I have outlined them below, referring to the line on the manuscript: [039]: "we report positioning precision..." This is awkwardly worded

We now write "We report a positioning precision".

In a number of places, the citation is placed after a word without a space between the last word and the reference number. Please add a space to make consistent throughout the text. I've referenced all the ones I could see below: [050,051,053,055,071,072,078,081,085,086,090,092,094,095,148,169,248,310,315,335,336]

We have added spaces before the reference number throughout the whole text.

The sentence in lines [101-104] is somewhat unclear. Perhaps structuring the sentence inversely would make it more clear to the reader.

We have modified the order of the sentence for clarity, now writing **"In a process we refer to as δ -electron irradiation, we register a 200 keV electron beam with 20 nm spot size to the center of a diamond nanopillar; the nanopillar is fabricated in CVD-grown diamond with a 53 nm-deep, δ -doped nitrogen layer. "**

[352]: hyphen missing in delta doping (delta-doping)

We have added a hyphen following the reviewer's comments.

[363]: Possible typo in the unit of 98 ppm*nm ?

We have changed ppm*nm \rightarrow ppm·nm, following the reviewer's comments.

Reviewer 4

The manuscript reports a controlled creation of NV color centers in prefabricated diamond nanopillars and characterization of their properties, primarily regarding nanoscale applications such as quantum sensing. It is a nice work built upon the expertise of the authors in diamond wafer growth, color center formation, nanofabrication, and quantum sensing. I would like to raise following questions and make some comments:

Main questions:

Q1. I am not sure if the 'estimated (not measured with a test field)' AC sensitivity of 42 nT/sqrt(Hz) (see fig5 and associated discussion) is a significant breakthrough. Because I would like to cite a few earlier works and want to understand what is (are) the main factor(s) limiting the sensitivity here. First, if I am not wrong, the NV centers in the studied sample here are surrounded by an isotopically pure (^{13}C depleted) carbon environment, and the previous works (e.g., Nature Materials, volume 8, page 383 (2009)) on the isotopically pure samples showed AC (Hahn) sensitivity of

4 nT/sqrt (Hz) per single ‘bulk’ NV. Second, in a non-isotopic sample, Grinolds et al (Nature Physics volume 9, page 215 (2013)), achieved 56 nT/sqrt(Hz) AC (Hahn) sensitivity per single NV in a nanopillar. So, I want to know despite having a nanopillar and a pure sample, why the sensitivity has not improved substantially?

We appreciate the reviewer’s question regarding the sensitivities presented in Fig. 5, and we are happy to clarify the improvements afforded by our technique. Indeed the median sensitivity value of 42 nT/Hz^{1/2} is not better than best reported sensitivities. However, the context is important here. The $\eta = 4$ nT/Hz^{1/2} reported in Nature Materials 8, 383 (2009), is achieved on *one NV deep* in an *unpatterned diamond*, and hence does not provide a robust and reliable path towards scalable production of sensitive and useful magnetometers that can be, for instance, incorporated into a scanning probe microscopy setup. The $\eta = 56$ nT/Hz^{1/2} reported in Nature Physics 9, 215 (2013) (demonstrated in a useful device) is a “hero” device that is state-of-the-art for shallow NVs in nanopillars and is consistent with the distribution of implanted NV centers we show in Fig. 5b. Our technique, as discussed in the text surrounding Fig. 5, features the reliable formation of NVs with improved median sensitivity over implanted and non- δ -e irradiated NV centers in a device-ready form factor. The reliably-formed 42 nT/ $\sqrt{\text{Hz}}$ median sensitivity of NVs in pillars should not be compared to the highest reported values. Rather, one should compare to the range of sensitivities from conventional methods, which we characterize as having a median of 121 nT/ $\sqrt{\text{Hz}}$ (see Fig. 5). Our method can yield sensitivities down to 20 nT/ $\sqrt{\text{Hz}}$, which exceeds the reported values for NVs in nanopillars that we are aware of.

We further note that our T₂, which is limited by the paramagnetic P1 environment and hence is shorter than the two above referenced papers, can be readily improved with lower nitrogen concentration. We used an intentionally high N doping to demonstrate tunability of NV number (Fig. 2). Our group has previously shown (e.g. Nano Letters 16, 2450 (2016)) that nitrogen δ -doping combined with electron irradiation with lower nitrogen doping can lead to significantly longer T₂’s in excess of 1 ms. We have also added text in the conclusion to this effect: **“Reducing the nitrogen density within the δ -doped layer is expected to extend T_2^{Hahn} beyond 1 ms [Nano Letters 16, 2450 (2016)]”**

We have also added a discussion to the main text regarding the highest achievable sensitivities for single NV scanning probes using our method. The following text has been added to the third subsection, i.e., High-yield, scalable magnetic field sensors, of the Main section: **For comparison, the reported values of η of single NV magnetometers for nanoscale imaging include 56 nT/ $\sqrt{\text{Hz}}$ [Nature Nanotechnology 7, 320–324 (2012)], 56 nT/ $\sqrt{\text{Hz}}$ (dynamical decoupling with 512-pulse XY8: 18 nT/ $\sqrt{\text{Hz}}$) [Nature Physics 9, 215–219 (2013)], 104 nT/ $\sqrt{\text{Hz}}$ [Nature Physics 17, 1216–1220 (2021)], and 66 nT/ $\sqrt{\text{Hz}}$ (dynamical decoupling with 128-pulse XY8: 32 nT/ $\sqrt{\text{Hz}}$) [Physical Review Applied 17, 054008 (2022)].**

Q2. Comparing with the work of Wang et al., Sci. Adv. 8, eabn9573 (2022) which the authors cite in this manuscript (reference 34), I think the results in the manuscript show significant enhancement of T₂ (Hahn) while relatively lower precision in NV

localization. So, with this trade off, can the authors highlight/envision what specific advantage their method achieves in quantum sensing over the performance (in quantum sensing) of devices reported in the ref. 34?

Indeed, Wang et al. show lateral positioning precision of ~ 15 nm, resulting in median PL_{sat} of 1.6 Mcps with 0.7 NA, and a median $T_2^{Hahn} \sim 10$ μ s. Magnetic field sensitivity $\eta \propto (PL_{sat} \times T_2^{Hahn})^{-0.5}$ (see Eq. 3 in the main text). Thus, our work with a ten-fold longer median T_2^{Hahn} significantly outweighs the 1.5-fold lower median PL_{sat} and yields sensors with better sensitivity. It should be noted that T_2^{Hahn} can be further enhanced beyond 1 ms by reducing the δ -doped nitrogen density, and we add the following text to the Conclusions: **“Reducing the nitrogen density within the δ -doped layer is expected to extend T_2^{Hahn} beyond 1 ms [*Nano Letters* 16, 2450 (2016)], and again, the tradeoff with NV number will need to be explored.”**

Moreover, as stated in the introduction, our NV formation technique is not limited to nanopillars and can be applied to a variety of nanophotonic devices, *e.g.* “bullseye” cavities [*Nano Letters* 15, 1493–1497 (2015)] to which a reference is now added in the conclusion. This versatility results because NV formation can be done after nanofabrication. In contrast, the work by Wang et al. cannot be easily extended to non-cylindrical structures, such as photonic crystals or resonators, as the diamond etch mask is defined by the implantation mask via undercut etching.

Another advantages of our technique is that, compared to Ref. 34, we can tune NV density to large values without incurring increased implantation damage that typically occurs when targeting dense ensembles.

Lastly, we note that the better localization afforded by Wang et al compared to our technique ($46 \rightarrow 15$ nm) does not improve nanopillar sensitivity much because PL_{sat} only changes by 10-15 % in the simulations of Wang et al, and even less in the simulations of our pillars shown in our supplement.

Q3. Is not the depth (~ 50 nm) of NV centers in this case is too deep for sensing nuclear spins on the diamond surface for doing molecular NMR/MRI? Also, is the predicted median AC sensitivity of 40 nT/sqrt (Hz) is sufficient to sense nuclear spins external to the diamond surface? For *e.g.*, if the field produced by an electron situated at 50 nm away (diamond surface) is 50 nT then the field produced by a single proton at similar distance will be in tens of pT, which requires far more sensitivity than the that is achieved (even with the dynamical decoupling sequences applied to NV centers studied here) in this work. So, in which area of quantum sensing/specific applications would their devices be advantageous? Can the authors make any such remarks about it in the discussion part?

We appreciate the reviewer’s comment regarding the applicability of our method to nanoscale NMR and we start by referencing our response to the first question of the Reviewer 2. Indeed a NV-sample distance of 50 nm is too deep for molecular NMR/MRI. However, though NV center-based nano-MRI is indeed an exciting frontier, there are many other NV-based sensing applications that do not rely on shallow enough NV centers for NMR/MRI. For example, NV centers located 20-100 nm below the diamond surface have been used to elucidate various nanomagnetic and nanoelectronic phenomena in a variety of condensed matter systems. Further, in scanning-NV

experiments that utilize diamond scanning probes for probing, *e.g.* magnetic materials, NV-sample distances of < 10 nm are in practice rarely achieved due to the difficulty of minimizing sensor-sample distance. This discrepancy between NV depth and NV-sample separation is well-known and discussed in the NV-scanning probe microscopy (NV-SPM) community and has recently been reported on - see Table 1 of ACS Nano 19 (8), 8255-8265 (2025)). Hence, for NV-SPM applied to condensed matter systems, a 50 nm NV depth is often more than sufficient, and further yields significantly more stable NV centers that retain their properties for long measurement times.

We further note that our method is completely compatible with the shallow (< 10 -nm-deep) NV centers that are desired for nanoMRI and we look forward to exploring δ -electron irradiated or δ -doped shallow NV centers in the future (or for others to apply our technique to shallow δ -doped NV centers). In response to the reviewer's question regarding applicability to nanoMRI, we have added the following sentence in the Conclusions: **“The versatility of our method makes it attractive to apply to, for instance, NV center formation in (111)-oriented diamonds [Physical Review X 15, 021035 (2025)] with prospects of improved photon count rates in nanopillars [Applied Physics Letters 104, 153108 (2014)], circular “bull’s-eye” cavities for enhanced photon collection [Nano Letters 15, 1493–1497 (2015)], as well as shallow ($< \sim 10$ nm deep) NV centers, with applications in NV-driven magnetic resonance imaging and ultra-high spatial resolution imaging of condensed matter systems.”**

Q4. Fig3c: There is a huge error bar for the smaller doses (esp. those close to 10^{19} e-/cm²) and the errors systematically reduce at higher doses. I assume this is simply because of the less probability of creating vacancies (and hence the NV complexes during annealing) for the low irradiation dose. Can the authors clarify this point and explain by adding more information in the text to help the reader?

Indeed the large error bar in Fig.3(c) at 4.8×10^{19} e-/cm² is due to the small number of NVs generated at this low dose. We have added the following sentence to the caption of Fig. 3(c): **“The error bars increase at lower doses due to the smaller number of measured NVs per spot”**

Q5. The spatial confinement (NV positioning precision in nanopillars) they are arriving at is based on the simulations since apparently the direct (experimental) measurements of this uncertainty is challenging due to influence of pillars' photonic modes. How reliable (in approximate percentage terms) is the method compared to measuring the positioning precision directly, say experimentally?

Our estimated value of lateral spatial confinement is based on simulations using the vacancy diffusion constant D_V measured from the NVs in the mesa region on the same sample. Thus the reliability of our estimation is limited by how precisely we measure D_V , resulting in an error of 1 nm and 2 nm for 280 nm and 480 nm pillars, respectively. A direct experimental measurement of spatial confinement in pillars is quite challenging to the best of our knowledge and requires advanced techniques such as scanning magnetic field gradients (as in Nature Physics 9, 215 (2013), PRL 113, 027602 (2014), and arxiv:2407.19576), which can reach ~ 5 nm, or nanoscale fluorescence quenching imaging (Nature Nano 7, 320 (2012)), which demonstrated 25.8 nm

precision. These methods require nanoscale scanning of the pillars in close proximity to a gradient-producing object, which can be cumbersome.

Other questions:

Q1. Given the deleterious effect of nanostructuring on the optical coherence/spectral properties of the NV centers, does this method also produce optically coherent NV centers for quantum networking applications? I know optimizing the optical coherence is probably not the main goal of this paper, but authors do seem to suggest their devices have generic quantum applications beyond the sensing.

We thank the reviewer for the question regarding the optical coherence of the NV centers created with our technique. We are actively working on answering this question and hope that others get interested as well. We are optimistic that our technique can achieve better optical coherence than conventional methods in two ways. First, our method does minimal damage to the host material with significantly fewer unwanted vacancy-related defects compared to implantation-based defect formation. It is known that these defects can serve as charge traps and lead to unwanted Stark shifts. Second, as stated by the reviewer, the damage due to the nanofabrication may induce additional noise. Our method allows positioning of the NVs away from the etched boundaries, which is typically not the mode maximum of the photonic modes, and thus suppresses the deleterious effects of nanofabrication.

Q2. The authors show 1.8 times enhancement of PL compared to non-irradiated pillars. Can the author comment on how does their absolute count rate compare (by taking into account the NA of their collection microscope objective) with the nanopillars created by other works for similar NV depth below the surface?

We have included the following text in the section on NV optical properties: **“Higher PL_{sat} values could be achieved with pillar geometries optimized for increased collection efficiency, such as parabolic pillars [*Physical Review Applied* 14, 064007 (2020)] or tapered pillars [*Science Advances* 8, eabn9573 (2022)], where with pillars with a $\sim 70^\circ$ sidewall angle, the authors achieved an average $PL_{sat} = 1.6$ Mcps.”** These results are amongst the best we are aware of for (100)-oriented diamonds with comparable NAs of ~ 0.7 .

Q3. Ref. 34 reports about 30% Rabi contrast while here the average Rabi contrast is 18%. What is the limiting factor here? Is it possible to further boost the Rabi contrast in these devices to 30%?

We thank the reviewer for this insightful question. We currently do not know what the limiting factor for Rabi contrast is in our devices. One possible explanation could be the short annealing time. We do note that the large Rabi contrast in Ref. 34 is quite impressive, though minimal information is given regarding the depth of the NVs measured for Fig. S6 in Ref 34.

Q4. Fig1c inset: what is the 532 nm laser power for obtaining the PL confocal scans of the nanopillars? Please include it in the caption at least.

We have added the following text to the caption of Fig. 1c: **“under 170 μ W of 532 nm excitation”**.

Q5. Page4: What is the need for doing air annealing at 450C, given that the NVs are fairly deep (>50 nm) below the surface? If this surface termination is not done, does the T2 get worse? If so, by what amount?

The air anneal is done as part of a standard protocol in our group, where we work with samples of a wide range of NV depths. Indeed, we have seen deleterious surface effects on even 50-nm-deep NVs in other samples, so as a precaution, we do our most thorough surface preparation on all of our samples. We did not do a systematic study on the effects of the air anneal for NVs in this study.

Q6. In the concluding part, the authors allude to improving the coherence in the future works but do not mention what specific potential strategies could be employed to increase the coherence (T_2) beyond their current 100 us range.

We have added the following sentence to the Conclusions: **“Reducing the nitrogen density within the δ -doped layer is expected to extend T_2^{Hahn} beyond 1 ms [*Nano Letters* 16, 2450 (2016)], and again, the tradeoff with NV number will need to be explored.**

Q7. Also in the concluding part, I feel the authors’ suggestions for gaining more control over number and the positioning accuracy of the defects in nanopillars is a bit vague. For example, how does one achieve “in-situ annealing”? Similarly, the authors state in the same paragraph, “More accurate positioning can be achieved by reducing the annealing time”—reduction by what amount and at the cost of what (NV creation yield, coherence etc.?)?

Some of it is already mentioned in the main text while discussing the results of fig3, it would also help the reader if little more details are added to the concluding section with some appropriate literature references.

We thank the reviewer’s question regarding clarifying potential strategies for gaining more control over the number and positioning precision. For the control over the number, the following reference has been included to refer to the meaning of in-situ annealing: *Optica* 6, 662–667 (2019). For control over the positioning precision, the previous sentence has been replaced to the following: **“Lateral confinement can be enhanced by reducing the annealing time t_{anneal} , i.e., $\sigma_{loc} \propto t_{anneal}^{0.5}$, though the tradeoffs with potentially reduced coherence and NV yield will need to be optimized [*Physical Review Materials* 3, 113802 (2019)].”**

Q8. How do authors verify that the NV centers they are studying are singles? Have they performed the standard antibunching measurements to prove this?

We did perform $g^{(2)}$ autocorrelation measurements to help identify single NVs presented in Figs. 4 and 5. The data have been added to the supplement and the following text has been added to the main text: **“, confirmed by second-order correlation measurements (see Supplementary Section 8) and hyperfine-resolved pulsed ESR measurements.”**

Q9. Page7, line 298-299, why is the multiplicative factor of 400 ns used to get the average number of photons?

400 ns is the readout duration per measurement shot and is used to convert the count rate ($0.5 \times PL_{sat}$) to a photon number per measurement shot. 400 ns was chosen to optimize spin readout fidelity and was used for all data presented in this work.

Some comments:

C1. Page6 and fig4a: Authors report a measured T_2 value of nearly 100 us from the nanopillars but do not clarify whether it is from the 280 nm diameter pillar or from the 480 nm nanopillar.

The T2 values are measured on both 280 nm and 480 nm diameter nanopillars, and we observed no diameter-dependence of T2, which is consistent with the T2 being P1-limited. We have added the following text to the caption of Fig. 4a: **“in both 280 nm and 480 nm diameter pillars”**.

C2. Potentially misleading: The statement which seems to be highlighted in the abstract is that the lateral localization precision is 46 nm. This number for precision is true but corresponds to 280 nm diameter pillars. However, one of the supposedly key findings of the paper, the PL saturation counts (PL_{sat} ; 1.8X) is quoted from the 480 nm diameter pillar for which the NV localization precision is actually 72 nm. Similarly, I think it is not clear whether the T2 value of ~ 100 us is measured on pillars with dia. 280 nm or with dia. 480 nm. Finally, the best value for sensitivity (fig5) was also estimated based on the 480 nm pillar. If 480 nm pillars are their best devices (in terms of achievable sensitivity) then I would state the localization precision corresponding to that pillar (72 nm, not 46 nm). Anyway, I can understand that the authors might be intended to quote the best values from their work, however, in that case it would be fair to explicitly quote the pillar diameter for the key findings (localization precision, PL_{sat} , and T2) in the abstract (and wherever possible) to avoid the ambiguity for the reader.

We thank the reviewer for this comment; we do not wish to be misleading and have made modifications to the text. Indeed, the best localization occurs for 280-nm-diameter pillars, while the PL enhancement is maximized for 480-nm diameter pillars (for reasons explained in answer to both Reviewer 1’s question 2 and Reviewer 3’s question about the weak dependence of PL_{sat} on σ_{loc} for 280-nm-diameter pillars). We have modified the abstract to specify the pillar diameter for the 46-nm-lateral confinement by adding **“280 nm-diameter”**. We have also removed the reference to 1.8x higher PL_{sat} in the abstract, now just saying **“higher average photoluminescence”**, and we have relegated a more detailed discussion of PL enhancement to the new text in Spin and optical properties of single NVs in nanopillars Section, where we write: **“This can be attributed to a higher-order waveguiding mode, TE01, strongly outcoupling photons from NVs close to the edge of 280 nm pillars. In contrast, for 480 nm pillars, NVs close to the edge are weakly coupled to the TE01 mode, while NVs close to the center are coupled to both the fundamental mode HE11 and the higher order mode HE12, thus exhibiting sharper PL dependence in NV lateral displacement.”** As noted in the response to the previous question, the T2 is limited by P1s and does not depend on pillar diameter.

C3. It would be better to clarify in the main text that the saturation counts shown in the fig4 are already accounted for the counts from the background.

We have added the following to the main text, subsection “Spin and optical properties of single NVs in nanopillars”: **“(with background PL subtracted)”**

C4. Page5, first line, text in the bracket. I think the confocal of unpatterned mesas are in ‘top right and bottom left’ of fig2 while their positions are rotated in fig1. Accordingly the text (in page5) needs to be corrected. Please maintain the constituency.

We thank the reviewer for catching this inconsistency. We have rotated the confocal images in Fig. 2 to correspond to the SEM images in Fig. 1.

C5. Wherever the supplementary material is quoted, it would be better to specifically mention which section and figures of the supplementary material so that the reader is appropriately directed.

We have addressed this comment in our revised version.

Additional changes

We have included a section “Data availability” with a URL to a public repository containing source data files for all figures.

References

- [1] Hughes, L. B. *et al.* Strongly Interacting, Two-Dimensional, Dipolar Spin Ensembles in (111)-Oriented Diamond. *Physical Review X* **15**, 021035 (2025). URL <https://link.aps.org/doi/10.1103/PhysRevX.15.021035>.